# EAKV: An Entropy-Driven Adaptive KV Compression Framework for Long Video Understanding

Hengrui Hu [1]   Jingyu Li [2 3]   Juntao Liang [1]   Guanyu Chen [4]   Lan Zhang [1]

## Abstract

Although Multimodal Large Language Models (MLLMs) have made remarkable progress, they still struggle with long-video understanding due to the massive memory footprint of KV Caches. Existing methods often resort to disjoint retrieval or attention-based reduction with a uniform layer-wise budget to achieve compression. However, these methods disrupt temporal continuity and ignore the varying information density across network layers. In this work, we reveal that memory allocation should mirror layer-wise semantic density, rather than adhering to a uniform budget. To this end, we introduce EAKV, a training-free entropy-driven adaptive KV compression framework that leverages attention entropy to adaptively allocate compression budgets, selectively preserving critical tokens while distilling redundant contexts into compact contextual anchors, thereby achieving granular memory allocation proportional to semantic density. Extensive experiments on various benchmarks demonstrate that EAKV surpasses existing methods across diverse model architectures and varying parameter scales, yielding improvements ranging from 0.6% to 6.5%.

## 1. Introduction

The field of Multimodal Large Language Models (MLLMs) (Lin et al., 2024a; OpenAI, 2023; Li et al., 2024a; Hurst et al., 2024; Team et al., 2024; Bai et al., 2025a) has witnessed a rapid transition from simple image understanding to reasoning over intricate visual narratives in long videos. However, due to its long contexts, long-video reasoning faces a critical bottleneck: the unbounded accumulation of KV states during generation. For instance, processing a 1-hour video at 1 FPS yields nearly 1 million tokens under mainstream visual encoders. In a GQA-based 7B parameter model, this single sequence inflates the KV cache to an astronomical approximately 140 GB under FP16 precision, completely overwhelming the VRAM limits of commodity hardware and presenting a severe bottleneck for scalable deployment.

While Input-side Visual Token Reduction methods (Bolya et al., 2023; Liu et al., 2024b; Xing et al., 2025) effectively accelerate encoding, they offer limited relief for memory surges during the generation phase. Consequently, Inference-phase KV Cache Compression methods (Hooper et al., 2024; Cai et al., 2025a; Zhang et al., 2023; Li et al., 2024b; He et al., 2024; Zhang et al., 2025c) have emerged as a solution. However, existing KV compression methods exhibit fundamental shortcomings in long-form video contexts: Static compression approaches (Yuan et al., 2025; Liu et al., 2023; Oren et al., 2024; Tang et al., 2024) fall short by applying rigid budget allocations derived from simplistic priors (e.g., "deeper layers require more compression"), which overlook the dynamic nature of video semantics and fail to adapt to fluctuating information density. Sliding window strategies (Di et al., 2025; Chen et al., 2026; Luo et al., 2026) induce severe semantic fragmentation, disrupting long-range dependencies.

Moreover, existing attention-based methods prioritize absolute magnitude over distributional shape, rendering them susceptible to the "attention sink" bias (Xiao et al., 2024). This reliance on absolute magnitude-based Top-K selection leads the model to erroneously preserve high-scoring but non-critical tokens while sacrificing low-scoring context that is indispensable for reasoning. Consequently, they fail to differentiate between sharp peaks, which are indicative of focused retrieval, and the broad, flat distributions that are essential for deliberative reasoning. To overcome this limitation, we propose the entropy of attention matrices as a robust, distribution-aware metric. Unlike raw attention score, entropy offers a position-agnostic measure of "seman-

[1]School of Computer Science and Technology, University of Science and Technology of China, Hefei, China [2]Institute of Artificial Intelligence, Hefei Comprehensive National Science Center, Hefei, China [3]State Key Lab. for Novel Software Technology, Nanjing University, P.R. China [4]School of the Gifted Young, University of Science and Technology of China, Hefei, China. Correspondence to: Jingyu Li <jingyuli@iai.ustc.edu.cn>.

*Proceedings of the 43rd International Conference on Machine Learning*, Seoul, South Korea. PMLR 306, 2026. Copyright 2026 by the author(s).

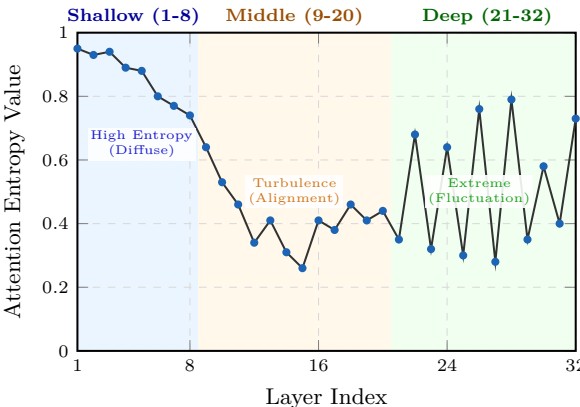

*Figure 1.* **Layer-wise Analysis of Attention Entropy in Qwen2.5-VL-7B model.** The distribution exhibits a non-monotonic pattern across network depths, which underscores the necessity for layer-wise adaptive budget allocation.

tic density" by prioritizing distribution shape over absolute magnitude, facilitating budget allocation that accurately reflects the model's cognitive state. Guided by this perspective, we conduct an extensive analysis of attention entropy in MLLMs, uncovering a critical "Semantic-Memory Mismatch" characterized by the following two key insights:

**Insight 1: High Spatio-Temporal Volatility.** Entropy variations within the same layer can reach a $4.2\times$ range, indicating that high-entropy regions with dense information need high-fidelity preservation, while low-entropy regions exhibit significant redundancy, permitting aggressive compression.

**Insight 2: Non-Monotonic Depth Dependency.** Contrary to the traditional view that information density decays monotonically with depth, we observe that attention entropy across MLLM layers exhibits complex, non-monotonic layer-wise patterns as shown in Figure 1. Shallow layers (1-8) exhibit high entropy to capture low-level textures, reflecting the broad, distributed attention required for initial contextual aggregation. Middle layers (9-20) exhibit a notable entropy reduction, corresponding to the dense cross-modal alignment between visual and textual features. Ultimately, deep layers (21-32) show significant fluctuations, driven by the dynamic context re-focusing necessary for complex reasoning rather than simple abstraction.

Building on these insights, we propose EAKV, an entropy-driven adaptive KV compression framework that shifts from rigid eviction to an "Information-Pay-As-You-Go" paradigm, establishing a dynamic mapping between local entropy and compression ratios: contexts with higher entropy are allocated larger KV budgets, while those with lower entropy are compressed into compact anchors, ensuring the allocation budget aligns precisely with semantic density. Our contributions can be summarized as follows:

- We conduct a pioneering study on attention entropy of MLLMs, revealing that information density is characterized by both depth-wise heterogeneity and high temporal volatility, underscoring the necessity of dynamic, adaptive budget allocation.

- We propose EAKV, a training-free framework that dynamically modulates KV budgets based on real-time entropy, preserving critical cues in high definition while distilling redundant contexts into compact anchors, thereby achieving granular memory allocation proportional to semantic density.

- Extensive experiments on various benchmarks demonstrate that EAKV surpasses existing methods across varying model scales with improvements ranging from 0.6% to 6.5%.

## 2. Related Work

### 2.1. Advancements in Long-Context MLLMs

The evolution of MLLMs has shifted from static image comprehension to long video understanding. Early models (Maaz et al., 2024; Lin et al., 2024a) relied on temporal pooling (Radford et al., 2021; Tschannen et al., 2025; Li et al., 2023) under fixed token budgets, while recent models (Bai et al., 2025a; Li et al., 2024a; Zhu et al., 2025) utilize dynamic resolution and scaled ViTs for fine-grained extraction. Despite extending context windows to millions of frames via advanced attention (Zhang et al., 2024) (Zhang et al., 2025a) and hierarchical long-context integration mechanisms (Zeng et al., 2026), the linear scaling of memory remains a bottleneck, rendering uncompressed long-video processing prohibitively expensive.

### 2.2. Visual Token Compression and Pruning

To alleviate the computational burden of extensive visual sequences, a substantial body of work focuses on reducing token redundancy before inference.

**Static and Heuristic Pruning.** Recognizing inherent spatio-temporal redundancy, FastV (Chen et al., 2024) and FitPrune (Ye et al., 2025b) employ attention-based pruning or early pooling within the vision encoder. Similarly, MADTP (Cao et al., 2024) optimizes token selection via training-free metrics to discard non-informative patches, streamlining the visual representation prior to LLM processing.

**Hierarchical and Adaptive Compression.** Beyond simple pooling, Chat-UniVi (Jin et al., 2024) and SlowFast-LLaVA (Xu et al., 2024) cluster tokens into semantic summaries to balance detail with breadth. Alternatively, PyramidDrop (Xing et al., 2025), TC-Former (Zeng et al., 2024a), and Kangaroo (Liu et al., 2024a) employ progressive token dropping or speculative decoding to retain salient regions. While

these input-side methods effectively reduce FLOPs, they incur irreversible information loss. Subtle but critical cues cannot be recovered if they are pruned early, which imposes a performance ceiling on video understanding.

## 2.3. KV Cache Management and Optimization.

Distinct from input reduction, inference-stage optimization targets the management of Key-Value (KV) caches, the primary memory bottleneck for long-context generation.

**General KV Cache Compression.** $H_2O$ (Zhang et al., 2023), SnapKV (Li et al., 2024b), and PyramidKV (Cai et al., 2025b) employ dynamic eviction policies that retain crucial KV pairs based on accumulated attention scores, while DuoAttention (Xiao et al., 2025) further offloads non-critical heads to constant-time access patterns.

**Video-Specific Memory Systems.** To manage temporal scale, existing works employ sliding windows (Song et al., 2024), summary vectors (Shu et al., 2025), or external retrieval-based offloading (Di et al., 2025). Similarly, cognitive-inspired frameworks (Zeng et al., 2024b) attempt to emulate human memory mechanisms for context management. However, these paradigms remain sub-optimal: segmentation induces temporal fragmentation that severs cross-boundary causal dependencies, while retrieval-based methods rely on surface similarity, overlooking implicit context vital for deliberative reasoning. Besides, cognitive-inspired frameworks inevitably introduce excessive computational overhead. Departing from the above methods, our method dynamically maps entropy to compression ratios, ensuring memory usage is strictly proportional to semantic density.

## 3. Methodology

The architecture of our proposed **EAKV** framework is illustrated in Figure 2. To distinguish essential context from redundancy, the Fine-Grained Entropy Valuation module quantifies information density at the specific attention head level. Building on this valuation, a Entropy-Driven Adaptive Allocation module dynamically distributes the memory budget based on entropy competition. Finally, Asymmetric Temporal Integration and Hybrid Token Compression module compress non-salient KV pairs, synthesizing them into contextual anchors to maintain global consistency. The general pipeline and detailed modules are elaborated below.

### 3.1. General Pipeline

Let $F, T, S, L$, and $D$ denote the number of frames, tokens per frame, prompt length, the number of layers, and the dimension of model, respectively. Each layer contains $H$ heads with dimension $d_k = D/H$. Given raw video and text inputs, the vision encoder and projection layer derive

visual features $\mathbf{X} \in \mathbb{R}^{F \times T \times D}$, while the embedding layer produces prompt features $\mathbf{P} \in \mathbb{R}^{S \times D}$. We flatten $\mathbf{X}$ to form a input visual sequence $\mathbf{Z} \in \mathbb{R}^{M \times D}$, where $M = F \cdot T$ is the total sequence length. We introduce a compression ratio $\rho \in (0, 1]$ to define this budget relative to the theoretical full cache size. As the MLLM processes $\mathbf{Z}$, we maintain a KV cache $\mathcal{C}$ within a fixed global memory budget $B_{total}$, where $B_{total} = \rho \cdot M \cdot L \cdot H$. Unlike conventional approaches that indiscriminately accumulate historical states, we introduce a selective compression function $\Phi$ to dynamically manage information density. As the sequence is processed in consecutive chunks, let $\mathbf{K}_i^{(l,h)}$ and $\mathbf{V}_i^{(l,h)}$ represent the key and value states generated by the $i$-th chunk at head $h$ of layer $l$. The cache update rule at step $i$ is formulated as:

$$\mathcal{C}_i^{(l,h)} = \Phi^{(l,h)} \left( \mathcal{C}_{i-1}^{(l,h)}, \mathbf{K}_i^{(l,h)}, \mathbf{V}_i^{(l,h)} \right). \quad (1)$$

Here, $\Phi^{(l,h)}$ integrates the incoming states with the existing historical cache $\mathcal{C}_{i-1}^{(l,h)}$. By selectively retaining essential KV pairs, we ensure global consistency across long sequences. The pseudocode for our algorithm is shown in Appendix A.

### 3.2. Fine-Grained Entropy Valuation

To evaluate content-based saliency independent of positional bias, we compute attention scores prior to the application of Rotary Positional Embeddings (RoPE) (Su et al., 2024). The attention matrix $\mathbf{A}^{(l,h)}$ for head $h$ in layer $l$ is:

$$\mathbf{A}^{(l,h)} = \text{Softmax} \left( \frac{\mathbf{Q}^{(l,h)}(\mathbf{K}^{(l,h)})^\top}{\sqrt{d_k}} \right) \in \mathbb{R}^{L_q \times L_k}, \quad (2)$$

where $L_q$ and $L_k$ denote the query and key sequence lengths, respectively. To quantify the information density of this attention head, we define the head-level entropy $E^{(l,h)}$ as the mean Shannon entropy across all queries:

$$E^{(l,h)} = -\frac{1}{L_q} \sum_{i=1}^{L_q} \sum_{j=1}^{L_k} A_{i,j}^{(l,h)} \log(A_{i,j}^{(l,h)} + \epsilon). \quad (3)$$

Higher entropy signifies a head aggregating global context with high semantic density, whereas lower entropy indicates a focus on specific, localized tokens.

### 3.3. Entropy-Driven Adaptive Allocation

Building upon this valuation, we propose a head-specific allocation mechanism that integrates a static *Layer-wise Safety Net* and a dynamic *Entropy-Driven Contention* to enforce the global budget $B_{total}$:

$$\sum_{l=1}^{L} \sum_{h=1}^{H} N^{(l,h)} \leq B_{total}, \quad (4)$$

where $N^{(l,h)}$ denotes the number of retained tokens for the $h$-th head in the $l$-th layer.

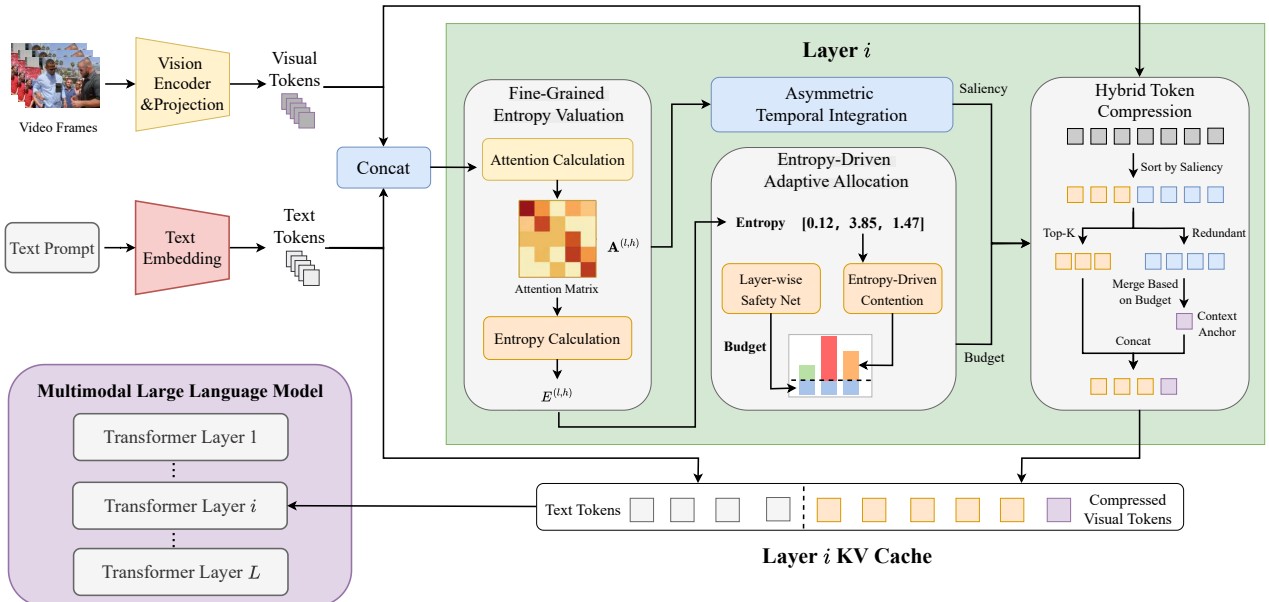

*Figure 2.* **The architecture of our proposed EAKV.** The pipeline begins with (1) Fine-Grained Entropy Valuation, which calculates the entropy metric to measure the information density of each attention head. This metric drives the (2) Entropy-Driven Allocation, integrating a Layer-wise Safety Net to ensure basic connectivity and an Entropy-Driven Contention strategy to prioritize heads with rich semantic content. Then (3) Asymmetric Temporal Integration stabilizes saliency scoring, enabling that (4) Hybrid Token Compression module to retain salient tokens and compresses background noise into contextual anchors guided by the assigned budget.

**Layer-wise Safety Net.** To guarantee minimal functional connectivity and prevent feature collapse in deep layers, we reserve a fixed ratio $\alpha$ of the total budget as the base capacity. This portion is distributed uniformly across all heads, ensuring that every attention head maintains a functional lower-bound receptive field regardless of its entropy:

$$N_{base}^{(l,h)} = \left\lfloor \frac{\alpha \cdot B_{total}}{L \cdot H} \right\rfloor. \qquad (5)$$

**Entropy-Driven Contention.** The remaining capacity, $B_{flex} = (1 - \alpha) \cdot B_{total}$, is dynamically distributed via a global contention mechanism. Specifically, we apply Z-score normalization to the raw head-wise entropy $E^{(l,h)}$ across all $L$ layers and $H$ heads to obtain a standardized metric $\hat{E}^{(l,h)}$:

$$\hat{E}^{(l,h)} = \frac{E^{(l,h)} - \mu_E}{\sigma_E}, \qquad (6)$$

where $\mu_E = \frac{1}{L \cdot H} \sum_{l'=1}^{L} \sum_{h'=1}^{H} E^{(l',h')}$ and $\sigma_E$ denote the global mean and standard deviation of the entropy values, respectively. We then derive the contention weight $w^{(l,h)}$ via a temperature-scaled Softmax:

$$w^{(l,h)} = \frac{\exp(\hat{E}^{(l,h)}/\tau)}{\sum_{l'=1}^{L} \sum_{h'=1}^{H} \exp(\hat{E}^{(l',h')}/\tau)}. \qquad (7)$$

Finally, the allocated KV budget for each head combines its static safety net with its earned dynamic portion:

$$N^{(l,h)} = N_{base}^{(l,h)} + \left\lfloor w^{(l,h)} \cdot B_{flex} \right\rfloor. \qquad (8)$$

This strategy reconciles structural integrity with semantic focus: While *Layer-wise Safety Net* safeguards basic connectivity across all layers, *Entropy-Driven Contention* allocates memory resources to heads with high semantic density. As proven in Appendix B, it theoretically optimizes KV cache information retention, maintaining layer-wise functionality while prioritizing complex semantic dependencies.

### 3.4. Asymmetric Temporal Integration

To ensure stable token ranking across streaming chunks, we introduce Asymmetric Temporal Integration, which employs "Fast-Attack, Slow-Decay" dynamics to decouple signal accumulation from dissipation, enhancing robustness against transient occlusions. Specifically, we update the cumulative saliency $\tilde{\mathbf{S}}^{(l,h)}$ by smoothing the instantaneous attention $\mathbf{S}_{inst}^{(l,h)}$ with a trajectory-aware coefficient $\lambda$:

$$\tilde{\mathbf{S}}_i^{(l,h)} = \lambda_i \mathbf{S}_{inst}^{(l,h)} + (1 - \lambda_i)\tilde{\mathbf{S}}_{i-1}^{(l,h)}, \qquad (9)$$

where $\lambda_i$ is adaptively triggered by the temporal trend:

$$\lambda_i = \begin{cases} \beta_{\text{rise}} & \text{if } \mathbf{S}_{inst}^{(l,h)} > \tilde{\mathbf{S}}_{i-1}^{(l,h)}, \\ \beta_{\text{decay}} & \text{otherwise.} \end{cases} \qquad (10)$$

By setting $\beta_{\text{rise}} > \beta_{\text{decay}}$, EAKV rapidly locks onto emerging visual cues while maintaining hysteresis for fading signals. As proven in Appendix C, this strategy effectively mitigates the "flickering eviction" of persistent objects, providing a stable foundation for subsequent compression.

### 3.5. Hybrid Token Compression

Leveraging the stable saliency provided by asymmetric integration, we identify the top-$k^{(l,h)}$ indices $\mathcal{I}_{top}$ and the residual set $\mathcal{I}_{rest}$, where $k = \max(N^{(l,h)} - 1, 0)$. To prevent information loss, we propose to aggregate residuals into a *Contextual Anchor* rather than indiscriminate eviction:

$$\mathbf{z}_K^{(l,h)} = \sum_{j \in \mathcal{I}_{rest}} \omega_j \mathbf{K}_j^{(l,h)}, \quad \mathbf{z}_V^{(l,h)} = \sum_{j \in \mathcal{I}_{rest}} \omega_j \mathbf{V}_j^{(l,h)}, \tag{11}$$

where $\omega_j$ represents the normalized importance weight. The final compressed output is:

$$\Phi^{(l,h)}(\mathbf{K}_i, \mathbf{V}_i) = \text{Concat}\left( (\mathbf{K}[\mathcal{I}_{top}], \mathbf{V}[\mathcal{I}_{top}]), (\mathbf{z}_K^{(l,h)}, \mathbf{z}_V^{(l,h)}) \right) \tag{12}$$

To preserve temporal causality and relative positional relationships, we retain original Position IDs for $\mathcal{I}_{top}$ and assign a "Center-of-Mass" Position ID for the context anchors as detailed in Appendix D. By doing so, EAKV preserves crisp details for salient entities while condensing background redundancy into a semantically pure representation.

## 4. Experiments

### 4.1. Benchmarks

We evaluate our approach on four representative video understanding benchmarks, covering diverse durations and task complexities:

**Video-MME** (Fu et al., 2025) features 900 videos (256 hours) and 2,700 annotated QA pairs across 30 subfields. It categorizes content into three duration tiers: short (<2 min), medium (4–15 min), and long (30–60 min).

**MLVU** (Zhou et al., 2025) covers nine tasks with video lengths ranging from 3 minutes to 2 hours, offering a broad spectrum of temporal scales.

**LongVideoBench** (Wu et al., 2024) contains 3,763 videos and 6,678 questions. It tests the model's ability to find specific, fine-grained details in videos that are up to one hour long.

**LVBench** (Wang et al., 2025b) focuses on ultra-long comprehension with an average duration of 4,101 seconds. Its 1,549 QA pairs assess complex capabilities including temporal grounding and key information retrieval.

### 4.2. Implementation

We incorporate EAKV into a suite of MLLMs to evaluate its versatility. Our experiments cover the LLaVA-Video, InternVL3.5, Qwen2-VL and Qwen2.5-VL series, encompassing the 3B, 7B, and 8B parameter variants. Videos are sampled at a rate of 2 fps and a maximum frame limit of 2048. The EAKV hyperparameters are empirically set to $\alpha = 0.3$, $\tau = 1.0$, $\beta_{\text{rise}} = 0.8$, and $\beta_{\text{decay}} = 0.1$. All evaluations are conducted using the lmms-eval framework (Zhang et al., 2025b).

### 4.3. Main Results

#### 4.3.1. COMPARATIVE PERFORMANCE ANALYSIS

**Comparison with Existing MLLMs.** As shown in Table 1, we evaluate our proposed EAKV framework against state-of-the-art proprietary and open-source models across four demanding long-video benchmarks. EAKV demonstrates consistent superiority across the 3B, 7B, and 8B model scales. Highlighting its scalability, the framework shows robust generalization on the compact 3B scale, improving Qwen2.5-VL by +4.1% on VideoMME (Long). When applied to Qwen2-VL-7B, it yields substantial gains, such as a +5.1% surge on MLVU and a prominent +6.5% leap on LVBench. EAKV also consistently upgrades LLaVA-Video-7B, yielding a +3.3% absolute improvement on VideoMME (All) and driving LVBench up to 47.4%. Furthermore, this consistent enhancement extends seamlessly to the 8B scale, where EAKV successfully improves InternVL3.5-8B across all benchmarks. Notably, this performance outperforms all baseline models, confirming that our framework effectively balances memory efficiency with semantic understanding capabilities. More results on Video-MMMU are provided in Appendix E.

**Comparison with Existing Compression Methods.** As summarized in Table 2, EAKV consistently outperforms state-of-the-art MLLM compression techniques across all evaluated benchmarks. To ensure a fair and rigorous comparison, all evaluated frameworks are implemented at the Qwen2.5-VL-3B model. Methods such as FastV, FitPrune, and SparseVLM use accumulated attention or recycling but overlook dynamic information density. Similarly, Pyramid-Drop employs a layer-wise monotonic decay for budget allocation but contradicts the non-monotonic layer importance observed in Section 1, leading to suboptimal retention. Although VL-Cache explores dynamic allocation via heuristics, our theoretically grounded, entropy-aware mechanism demonstrates superior efficacy and achieves 46.7% on LVBench and 66.8% on MLVU, surpassing the strongest baselines by 2.8% and 0.5%, respectively.

*Table 1.* **Performance comparison with existing MLLMs on four benchmarks.** EAKV consistently improves the host models at 3B, 7B, and 8B scales. Notably, with Qwen2.5-VL-7B, EAKV achieves state-of-the-art performance among open-source models, delivering a 4.8% improvement on the challenging LVBench benchmark.

| Model | Size | VideoMME | | LongVideoBench | MLVU | LVBench |
|---|---|---|---|---|---|---|
| | | All | Long | Val | Dev | Val |
| **Proprietary Vision-Language Models** | | | | | | |
| GPT-4o (Hurst et al., 2024) | – | 71.9 | 65.3 | 66.7 | – | 30.8 |
| Gemini-1.5-Pro (Team et al., 2024) | – | 75.0 | 67.4 | 64.0 | – | 33.1 |
| **Open-Source Vision-Language Models** | | | | | | |
| VILA-1.5 (Lin et al., 2024b) | 3B | 59.7 | 50.1 | 53.2 | 60.3 | 42.5 |
| InternVideo2.5 (Wang et al., 2025c) | 4.2B | 61.0 | 51.6 | 55.1 | 64.7 | 44.2 |
| BLIP-3-Video (Ryoo et al., 2025) | 3.9B | 61.2 | 53.9 | 55.7 | 62.8 | 44.2 |
| Qwen2.5-VL (Bai et al., 2025b) | 3B | 61.5 | 51.2 | 54.2 | 64.8 | 43.3 |
| **Qwen2.5-VL + Ours** | **3B** | **63.0** | **55.3** | **56.4** | **66.8** | **46.7** |
| mPLUG-Owl3 (Ye et al., 2025a) | 7B | 59.3 | 50.1 | 52.1 | 63.7 | – |
| NVILA (Liu et al., 2025) | 8B | 64.2 | 54.8 | 57.7 | 70.1 | – |
| Video-LLaMA3 (Zhang et al., 2025a) | 7B | 66.2 | 54.9 | 59.8 | 73.0 | 45.3 |
| ViLAMP (Cheng et al., 2025) | 7B | 67.5 | 57.8 | 60.2 | 70.2 | 46.3 |
| LLaVA-Video (Zhang et al., 2025d) | 7B | 62.9 | 52.4 | 58.2 | 67.6 | 44.2 |
| **LLaVA-Video + Ours** | **7B** | **66.2** | **55.1** | **59.9** | **69.3** | **47.4** |
| InternVL3.5 (Wang et al., 2025a) | 8B | 64.6 | 55.6 | 62.1 | 65.5 | 41.8 |
| **InternVL3.5 + Ours** | **8B** | **65.9** | **56.2** | **63.8** | **66.7** | **45.2** |
| Qwen2-VL (Wang et al., 2024) | 7B | 63.3 | 53.8 | 55.6 | 66.9 | 42.4 |
| **Qwen2-VL + Ours** | **7B** | **64.5** | **55.7** | **57.2** | **72.0** | **48.9** |
| Qwen2.5-VL (Bai et al., 2025b) | 7B | 65.4 | 55.6 | 59.5 | 70.2 | 45.3 |
| **Qwen2.5-VL + Ours** | **7B** | **67.8** | **59.2** | **63.3** | **74.8** | **50.1** |

*Table 2.* **Comparison with Existing Compression Methods.** EAKV consistently outperforms static (e.g., FastV) and heuristic (e.g., VL-Cache) strategies.

| Method | VideoMME | | MLVU | LVBench |
|---|---|---|---|---|
| | Long | All | Dev | Val |
| ∞-Video (Santos et al., 2025) | 38.9 | 42.4 | – | – |
| LongVU (Shen et al., 2024) | 53.9 | 60.1 | 64.0 | – |
| PyramidDrop (Xing et al., 2025) | 53.1 | 60.5 | 63.7 | 41.6 |
| SparseVLM (Zhang et al., 2025c) | 54.4 | 60.7 | 63.0 | 43.9 |
| Video-X$^2$L (Qin et al., 2025) | 53.8 | 60.9 | 66.3 | – |
| LOOK-M (Wan et al., 2024) | 53.6 | 61.0 | 63.8 | 42.6 |
| FastV (Chen et al., 2024) | 53.5 | 61.2 | 63.2 | 42.3 |
| FitPrune (Ye et al., 2025b) | 53.6 | 61.2 | 63.6 | 42.0 |
| VL-Cache (Tu et al., 2025) | 53.2 | 61.3 | 64.5 | 42.4 |
| **Ours** | **55.3** | **63.0** | **66.8** | **46.7** |

outperforms the baseline by a substantial margin. By consistently achieving near or above 87% reduction rates in long-context scenarios (e.g., 91.3% on LongVideoBench), EAKV pushes the boundaries of token reduction while strictly preserving the semantic integrity.

Crucially, this reduction translates directly into expanded context capacity. As highlighted in Figure 3, EAKV effectively breaks memory bottlenecks for long-form videos. Under identical memory constraints, our method significantly increases the supported frame count across all benchmarks, culminating in up to 2.1× more frames than the baseline and a remarkable 11.5× effective context expansion on LongVideoBench. This capacity upgrade empowers the model to reason over long-range temporal dependencies that are inaccessible to existing methods.

### 4.3.3. COMPUTATIONAL EFFICIENCY AND MEMORY FOOTPRINT

We conduct a comprehensive profiling of the computational and memory overhead introduced by our method, proving that our method effectively breaks the memory bottleneck of ultra-long video inference with negligible extra cost.

### 4.3.2. COMPRESSION EFFICIENCY AND CONTEXT EXPANSION

We evaluate compression efficiency across varying temporal durations, specifically targeting the redundancy-rich "Long" category. As shown in Table 3, our entropy-driven strategy

*Table 3.* **Comparison of Token Reduction Rates and Compression Ratios.** EAKV consistently outperforms the baseline(VL-Cache), achieving up to **11.5×** **compression**.

| Benchmark | Cat. | Baseline | Ours | | Gain |
|---|---|---|---|---|---|
| | | Red. Rate (%) | Red. Rate (%) | Comp. Ratio (×) | |
| Video-MME | Short | 55.4 | 63.2 | 2.7× | +7.8 |
| | Medium | 71.2 | 79.7 | 4.9× | +8.5 |
| | Long | 78.5 | **87.7** | **8.1×** | +9.2 |
| MLVU | Overall | 64.8 | 72.2 | 3.6× | +7.4 |
| | Long | 76.3 | **86.8** | **7.6×** | +10.5 |
| LongVideo | Overall | 69.5 | 78.4 | 4.6× | +8.9 |
| | *Long* | 81.5 | **91.3** | **11.5×** | **+9.8** |
| LVBench | Overall | 67.2 | 75.3 | 4.0× | +8.1 |
| | Long | 79.6 | **89.1** | **9.2×** | +9.5 |

**Note:** *Cat.*: Category; *Red. Rate*: Reduction Rate; *Comp. Ratio*: Compression Ratio; LongVideo is an abbreviation for LongVideoBench.

*Table 4.* **Average latency comparison on Qwen2.5-VL-7B.** TTFT denotes Time To First Token, and TPOT denotes Time Per Output Token.

| Phase | Metric | Baseline | Ours | Difference |
|---|---|---|---|---|
| **Prefill** | TTFT | 3.12 s | 3.28 s | +0.16 s |
| **Decode** | TPOT | 48.5 ms/token | 15.2 ms/token | **-33.3 ms/token** |
| **Overall** | Total | 15.54 s | 7.17 s | **-8.37 s** |

**Latency Analysis.** Computing the attention entropy $E^{(l,h)}$ occurs exclusively during the prefill phase. Since it is calculated directly on the already-materialized attention matrix $\mathbf{A}^{(l,h)}$, it introduces only a marginal $\mathcal{O}(H \cdot L \cdot T)$ overhead per head. This is negligible compared to the standard $\mathcal{O}(H \cdot L \cdot T^2)$ attention complexity. To validate this, we profile the Qwen2.5-VL-7B model on a single A100 GPU (processing 2048 frames and generating 256 tokens). The latency comparison is presented in Table 4. As shown, the prefill overhead from the entropy calculation is a negligible 5.1% (+0.16 s). However, by drastically reducing the KV cache size, our method significantly alleviates the memory bottleneck during the Decode phase, yielding a 3.2× speedup in token generation. Ultimately, the average total latency is reduced by 53.9%.

**Peak GPU Memory Consumption.** Absolute physical memory usage is a highly intuitive indicator of model efficiency. We profile the peak GPU memory consumption on a single 80GB A100 GPU across different models and frame counts. As summarized in Table 5, standard baselines exhibit catastrophic memory growth as the number of frames increases, leading to immediate Out-of-Memory

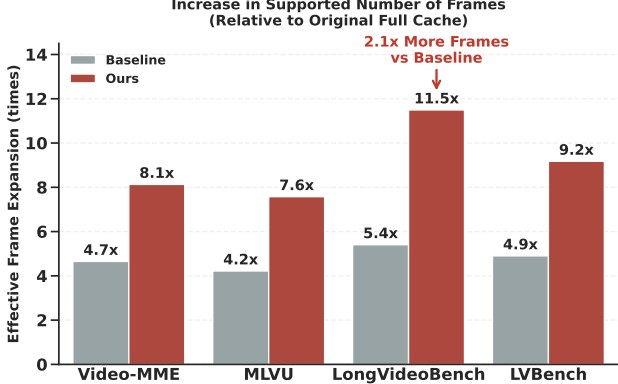

*Figure 3.* **Effective Context Expansion Analysis.** Compared to the baseline (VL-Cache), our method significantly extends the effective context length across all benchmarks. Notably on LongVideoBench, we achieve an **11.5×** **expansion** in frame capacity, supporting **2.1×** more frames than the baseline under the same memory constraint.

*Table 5.* **Peak GPU memory consumption (GB) on a single 80GB A100 GPU.** "OOM" indicates Out-of-Memory errors. Our method successfully eliminates OOMs and scales to 2048 frames.

| Model | Method | 256 Fr. | 512 Fr. | 1024 Fr. | 2048 Fr. |
|---|---|---|---|---|---|
| **Qwen2.5-VL-7B** | Baseline | 23.5 | 36.8 | 65.4 | OOM |
| | **Ours** | **18.4** | **23.8** | **33.5** | **51.6** |
| **LLaVA-Video-7B** | Baseline | 26.2 | 42.5 | OOM | OOM |
| | **Ours** | **19.6** | **25.5** | **36.4** | **55.8** |
| **InternVL-3.5-4B** | Baseline | 18.9 | 28.4 | 44.6 | 63.7 |
| | **Ours** | **16.1** | **22.7** | **32.9** | **43.8** |
| **Qwen3-VL-4B** | Baseline | 20.7 | 31.6 | 50.9 | 72.4 |
| | **Ours** | **17.5** | **24.9** | **36.8** | **48.2** |

(OOM) errors at 1024 or 2048 frames, particularly for 7B-parameter models. In contrast, our method significantly mitigates this memory growth curve. At an extreme context of 2048 frames, our method successfully bounds the footprint of 7B-parameter models to 51.6–55.8 GB (completely eliminating OOMs), and limits 4B-parameter models to 43.8–48.2 GB. This substantial reduction enables ultra-long video inference seamlessly on a single commercial GPU.

### 4.4. Ablation Studies

We conduct a comprehensive ablation study on Qwen2.5-VL-7B to validate the contribution of each component. The results summarized in Table 6 highlight two core advantages of our framework: precise semantic retention and unlocked scalability. Extended ablation analyses examining the Contextual Anchor and various hyperparameter configurations are provided in Appendix F and Appendix G, respectively.

*Table 6.* **Ablation study on different components on the Qwen2.5-VL-7B backbone.** The initial row represents the Qwen2.5-VL-7B baseline, while the final row corresponds to our complete method. The results demonstrate that our framework not only consistently boosts semantic reasoning at restricted context lengths (4K), but ultimately unlocks extreme temporal scaling (2048 frames) to achieve peak performance across all long-video benchmarks.

| Components | | | Max Frames | Context Length | VideoMME | LongVideoBench | MLVU | LVBench | $\Delta_{avg}$ |
|---|---|---|---|---|---|---|---|---|---|
| EDC | ATI | LSN | | | | | | | |
| | | | 128 | 8K | 65.4 | 59.5 | 70.2 | 45.3 | - |
| ✓ | | | 128 | 4K | 66.2 | 60.8 | 71.5 | 46.8 | +1.2 |
| ✓ | ✓ | | 256 | 4K | 67.1 | 62.1 | 72.8 | 48.2 | +1.2 |
| ✓ | ✓ | ✓ | 256 | 4K | 67.5 | 62.8 | 73.5 | 49.0 | +0.7 |
| ✓ | ✓ | ✓ | **2048** | **16K** | **67.8** | **63.3** | **74.8** | **50.1** | **+0.8** |

*EDC: Entropy-Driven Contention; ATI: Asymmetric Temporal Integration; LSN: Layer-wise Safety Net. $\Delta_{avg}$ denotes the average absolute performance improvement over the preceding row.*

**Efficiency over Quantity.** Implementing Entropy-Driven Contention yields an immediate gain ($\Delta_{avg} + 1.2\%$) despite reducing the cache context length from 8K to 4K. This counter-intuitive result confirms that our entropy metric effectively filters noise and proves that maintaining high semantic density is more critical for reasoning than simply retaining a longer, uncompressed history populated with redundant tokens.

**Structural & Temporal Optimization.** Asymmetric Temporal Integration efficiently aggregates temporal redundancy, allowing us to double the frame input $128 \rightarrow 256$ without expanding the memory footprint or compromising performance. The subsequent addition of the Layer-wise Safety Net further boosts stability ($\Delta_{avg} + 0.7\%$) by preventing feature collapse in deep layers, ensuring that aggressive compression does not disrupt sequential reasoning.

**Scaling to Extreme Contexts.** Crucially, the cumulative memory savings empower us to scale the input from 128 frames to an unprecedented 2048 frames, extending the context length to 16K. This capability unlocks a further performance surge ($\Delta_{avg} + 0.8\%$) and yields the best overall results (e.g., VideoMME reaches 67.8), demonstrating our framework successfully trades static memory overhead for massive temporal coverage, effectively solving the semantic-memory mismatch in long-video understanding.

### 4.5. Case Study

The visualization of retained tokens in Figure 4 reveals a striking distinction between attention-based and entropy-driven methods, illustrating how our method effectively overcomes the inherent constraints of attention. Attention-based methods attenuate sparse yet pivotal signals, such as the subtle temporal dynamics in Case 1 and the fine-grained spatial entities in Case 2, thereby marginalizing the critical cues required for precise reasoning. In contrast, our

approach leverages entropy as a proxy for semantic density, enabling a more nuanced and adaptive token compression strategy:

**Capturing Temporal Transients (Fig. 4a).** While the baseline smooths out subtle "head lowering" motions due to their low visual prominence, our method detects the surge in entropy generated by these temporal changes. By selectively preserving these motion-rich tokens, the model maintains the dynamic continuity essential for action recognition.

**Preserving Spatial Intricacies (Fig. 4b).** Where standard attention misinterprets fine-grained textures (e.g., company logos) as redundancy, our approach identifies their high information density. By focusing on regions with peak semantic density and ignoring redundant sinks, our strategy ensures that the model's reasoning is anchored in precise visual evidence, leading to reduced hallucinations.

## 5. Conclusion

This paper addresses the critical memory bottleneck in long-video understanding by challenging the effectiveness of static KV compression strategies. We identify a fundamental "Semantic-Memory Mismatch", demonstrating via attention entropy that information density is highly volatile and exhibits non-monotonic patterns across network layers. To resolve this, we propose EAKV, a training-free framework that transitions from rigid memory budgeting to a density-aware allocation paradigm. By dynamically preserving critical tokens based on entropy and synthesizing redundant contexts into compact anchors, EAKV achieves a precise alignment between memory usage and semantic importance. Extensive experiments confirm the efficacy of this approach, showing that EAKV consistently outperforms state-of-the-art methods with performance gains ranging from 0.6% to 6.5%.

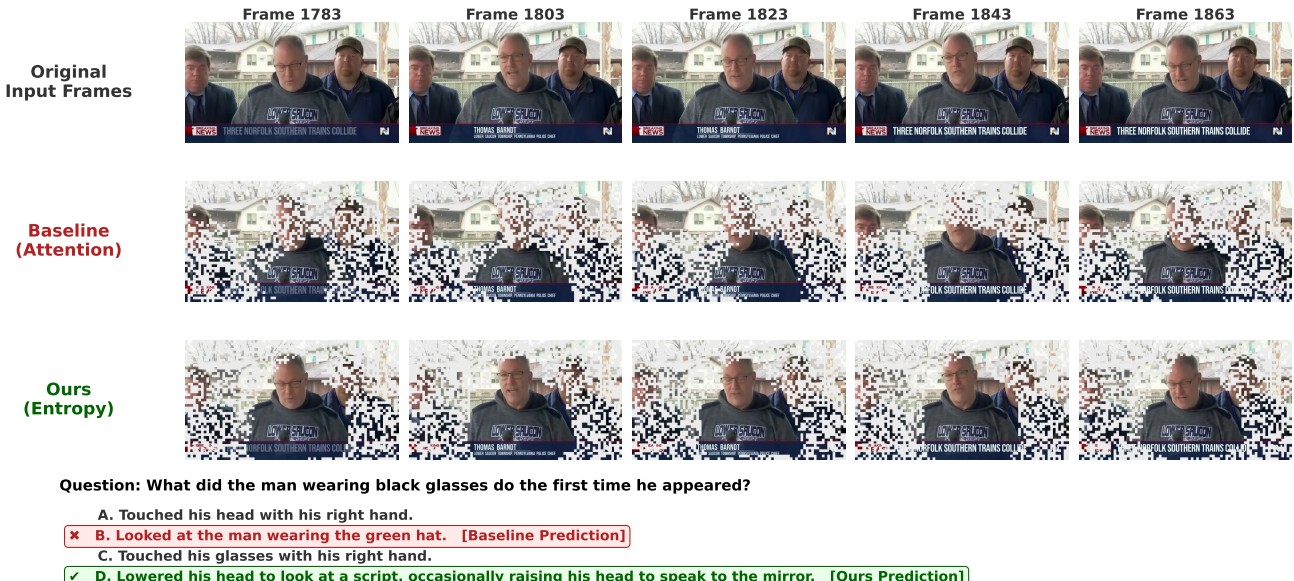

Question: What did the man wearing black glasses do the first time he appeared?

 A. Touched his head with his right hand.
 ✗ B. Looked at the man wearing the green hat. [Baseline Prediction]
 C. Touched his glasses with his right hand.
 ✓ D. Lowered his head to look at a script, occasionally raising his head to speak to the mirror. [Ours Prediction]

*(a)* **Dynamic Action.** The baseline overlooks subtle temporal cues like head lowering, leading to hallucination. Our method preserves high-entropy facial dynamics, enabling correct action recognition.

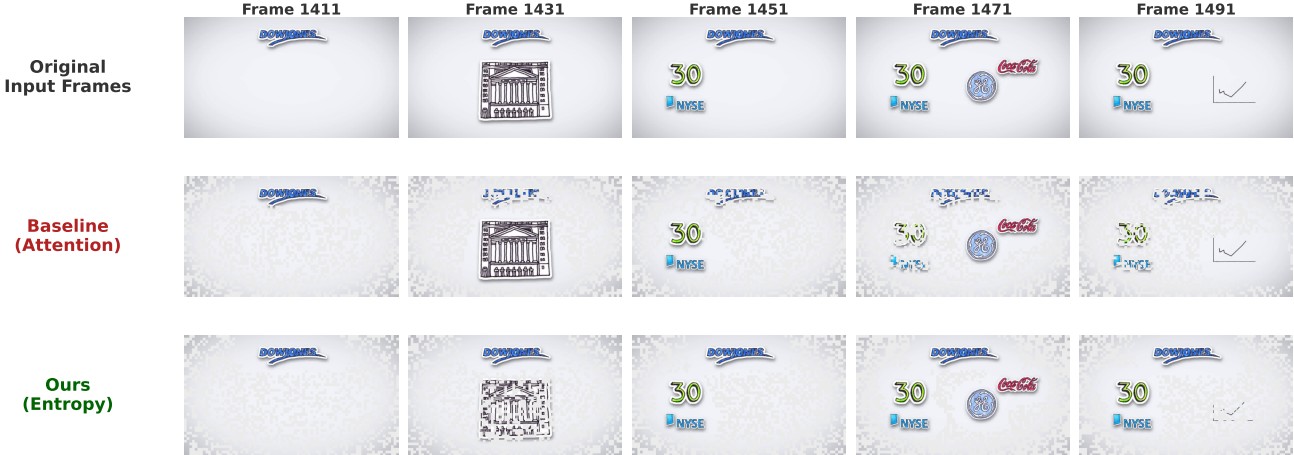

Question: What visual element is used to showcase companies that are included in the Dow Jones Industrial Average?

 A. Photographs of corporate buildings.
 B. Animated stock tickers scrolling across the screen.
 ✓ C. Hand-drawn company logos. [Ours Prediction]
 ✗ D. Charts showing stock performance over time. [Baseline Prediction]

*(b)* **Fine-grained Detail.** The baseline discards specific visual evidence like logos as redundancy. Our approach retains these high-complexity textures, supporting accurate entity identification.

*Figure 4.* **Qualitative comparison of visual token preservation.** We visualize the information retention differences between the attention-based baseline and our Entropy-Driven method. As observed, the baseline tends to discard semantically critical regions (e.g., facial expressions in (a) and text details in (b)) by treating them as background noise. Conversely, our method uses entropy to effectively distinguish signal from noise, ensuring that critical visual tokens are preserved. This ensures grounded reasoning and mitigates hallucinations.

## Limitations

While EAKV functions as a versatile plug-and-play framework compatible with diverse MLLMs, it operates strictly as an efficiency optimization layer rather than a semantic corrective. Consequently, even with the high-fidelity preservation of critical tokens, our framework cannot fix inherent generative flaws like hallucinations that originate from the backbone model.

## Acknowledgments

This work was supported in part by the China Postdoctoral Science Foundation (2025M771515), Major Industrial Innovation Plan Project of Anhui Provincial Development and Reform Commission (AHZDCYCXJH-QC2025-10) and Anhui Postdoctoral Scientific Research Program Foundation (2025C1166). The computational work in this paper was supported by the technical assistance of the Network Information Center and the Smart Campus Project at the University of Science and Technology of China. We gratefully acknowledge their support.

## Impact Statement

This work compresses the KV cache to improve the inference efficiency of Multimodal Large Language Models (MLLMs). By reducing energy consumption and memory requirements, our method contributes to "Green AI" and democratizes access to powerful models on consumer-grade hardware. We foresee no specific negative societal impacts beyond the general risks of LLM deployment.

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

# A. Algorithm Pseudo-code

Algorithm 1 details the execution flow of our Entropy-Driven Adaptive KV Compression Framework. It mathematically formalizes the interaction between the entropy-based budget allocation and the asymmetric temporal integration.

---

**Algorithm 1** EAKV: Entropy-Driven Adaptive KV Compression

---

1: **Input:** Token stream $\mathcal{X}$, Global Ratio $\rho$, Temp $\tau$, Safety $\alpha$, Decay $\{\beta_{rise}, \beta_{decay}\}$, Prompt length $P$.
2: **Init:** Cache $\mathcal{C}^{(l,h)} \leftarrow \emptyset$, Saliency $\tilde{\mathbf{S}}^{(l,h)} \leftarrow \mathbf{0}$ for layer $l \in [1, L]$, head $h \in [1, H]$
3: **for** each chunk $\mathbf{X}_t$ of length $M$ in $\mathcal{X}$ **do**
4:     $\mathbf{Q}, \mathbf{K}, \mathbf{V}, \mathbf{Pos} \leftarrow \text{ModelForward}(\mathbf{X}_t)$
5:     **PHASE 1: FINE-GRAINED ENTROPY VALUATION**
6:     **for** layer $l \in [1, L]$ and head $h \in [1, H]$ **do**
7:         $\mathbf{A}^{(l,h)} \leftarrow \text{Softmax}(\mathbf{Q}^{(l,h)}\mathbf{K}^{(l,h)^\top}/\sqrt{d_k})$
8:         $E^{(l,h)} \leftarrow -\frac{1}{P}\sum_{q=1}^{P}\sum_{k=1}^{M}\mathbf{A}_{q,k}^{(l,h)}\log(\mathbf{A}_{q,k}^{(l,h)} + \epsilon)$               // Head-level entropy
9:         $\mathbf{S}_{inst}^{(l,h)} \leftarrow \sum_{q=1}^{P}\mathbf{A}_{q,:}^{(l,h)}$                // Instantaneous saliency vector
10:    **end for**
11:    **PHASE 2: ENTROPY-DRIVEN ADAPTIVE ALLOCATION**
12:    $\hat{E}^{(l,h)} \leftarrow \text{Z-Score}(E^{(l,h)})$ across all $L \times H$ heads
13:    $w^{(l,h)} \leftarrow \frac{\exp(\hat{E}^{(l,h)}/\tau)}{\sum_{l',h'}\exp(\hat{E}^{(l',h')}/\tau)}$               // Global contention weight
14:    **for** layer $l \in [1, L]$ and head $h \in [1, H]$ **do**
15:        $N_{base} \leftarrow \lfloor \alpha \cdot \rho \cdot M \rfloor$               // Layer-wise Safety Net
16:        $N_{flex}^{(l,h)} \leftarrow \lfloor (1-\alpha) \cdot \rho \cdot M \cdot (L \cdot H) \cdot w^{(l,h)} \rfloor$               // Entropy-driven portion
17:        $N^{(l,h)} \leftarrow N_{base} + N_{flex}^{(l,h)}$               // Final assigned budget
18:    **end for**
19:    **PHASE 3: ASYMMETRIC INTEGRATION & HYBRID COMPRESSION**
20:    **for** layer $l \in [1, L]$ and head $h \in [1, H]$ **do**
21:        **for** $i \in [1, N]$ **do**
22:            $\lambda_i \leftarrow \begin{cases} \beta_{rise}, & \text{if } \mathcal{S}_{inst,i}^{(l,h)} > \tilde{\mathcal{S}}_i^{(l,h)} \\ \beta_{decay}, & \text{otherwise} \end{cases}$
23:            $\tilde{\mathcal{S}}_i^{(l,h)} \leftarrow \lambda_i \cdot \mathcal{S}_{inst,i}^{(l,h)} + (1-\lambda_i) \cdot \tilde{\mathcal{S}}_i^{(l,h)}$               // Asymmetric Temporal Integration
24:        **end for**
25:        $\mathcal{I}_{top} \leftarrow \text{TopK\_Indices}(\tilde{\mathbf{S}}^{(l,h)}, \max(N^{(l,h)} - 1, 0))$
26:        $\mathcal{I}_{rest} \leftarrow \{1, 2, \ldots, M\} \setminus \mathcal{I}_{top}$               // Gather all unselected residual tokens
27:        $\omega_i \leftarrow \frac{\tilde{\mathcal{S}}_i^{(l,h)}}{\sum_{j \in \mathcal{I}_{rest}} \tilde{\mathcal{S}}_j^{(l,h)} + \epsilon}, \quad \forall i \in \mathcal{I}_{rest}$
28:        $\tilde{\mathbf{K}}_{anc} \leftarrow \sum_{i \in \mathcal{I}_{rest}} \omega_i \mathbf{K}_i^{(l,h)}, \quad \tilde{\mathbf{V}}_{anc} \leftarrow \sum_{i \in \mathcal{I}_{rest}} \omega_i \mathbf{V}_i^{(l,h)}$
29:        $\mathbf{P}_{anc} \leftarrow \text{Round}(\sum_{i \in \mathcal{I}_{rest}} \omega_i \mathbf{Pos}_i)$               // Cache Update with Position IDs
30:        $\tilde{\mathbf{K}}_{out} \leftarrow \text{Concat}(\mathbf{K}^{(l,h)}[\mathcal{I}_{top}], \tilde{\mathbf{K}}_{anc}), \quad \tilde{\mathbf{V}}_{out} \leftarrow \text{Concat}(\mathbf{V}^{(l,h)}[\mathcal{I}_{top}], \tilde{\mathbf{V}}_{anc})$
31:        $\tilde{\mathbf{P}}_{out} \leftarrow \text{Concat}(\mathbf{Pos}[\mathcal{I}_{top}], \mathbf{P}_{anc})$
32:        $\mathcal{C}^{(l,h)} \leftarrow \text{Concat}(\mathcal{C}^{(l,h)}, \tilde{\mathbf{K}}_{out}, \tilde{\mathbf{V}}_{out}, \tilde{\mathbf{P}}_{out})$
33:    **end for**
34: **end for**

---

# B. Theoretical Analysis of Global Optimality with Entropy-Aware Allocation

We formally prove that the Entropy-Aware KV (EAKV) allocation strategy strictly minimizes the global approximation error compared to uniform allocation under a fixed memory budget.

## B.1. Formulation and Assumptions

**Definition B.1** (Global Optimization Problem). Let $\mathcal{L} = \{1, \dots, L\}$ be the set of layers. Let $k_l \in \mathbb{R}^+$ denote the KV budget for layer $l$, and $H_l$ denote the attention entropy. The global error minimization problem under a total budget constraint $K_{total}$ is:

$$\min_{\mathbf{k}} \mathcal{J}(\mathbf{k}) = \sum_{l=1}^{L} \mathcal{E}(H_l, k_l) \quad \text{s.t.} \quad \sum_{l=1}^{L} k_l = K_{total} \tag{13}$$

where $\mathbf{k} = [k_1, \dots, k_L]^\top$ is the allocation vector.

**Assumption B.2** (Error-Entropy Convexity). The layer-wise approximation error $\mathcal{E}(H_l, k_l)$ satisfies:

1. **Entropy Monotonicity:** $\frac{\partial \mathcal{E}}{\partial H_l} > 0$. Higher entropy implies a flatter distribution, leading to higher truncation error.

2. **Strict Convexity w.r.t Budget:** $\frac{\partial \mathcal{E}}{\partial k_l} < 0$ and $\frac{\partial^2 \mathcal{E}}{\partial k_l^2} > 0$. This reflects the diminishing marginal utility of retaining additional tokens.

3. **Separability:** $\mathcal{E}(H_l, k_l)$ can be modeled as $H_l \cdot g(k_l)$, where $g(\cdot)$ is a strictly convex, monotonically decreasing function. This assumes the truncation error is linearly proportional to the layer's entropy scale $H_l$, as higher entropy implies a flatter attention distribution where dropping tokens incurs a proportionately larger loss of information.

## B.2. Proof of Optimality

**Lemma B.3** (Optimal Allocation Condition). *The optimal budget allocation $\mathbf{k}^*$ satisfies $k_l^* \propto H_l$. Specifically, layers with higher entropy are allocated larger budgets.*

*Proof.* We construct the Lagrangian function with multiplier $\lambda$:

$$L(\mathbf{k}, \lambda) = \sum_{l=1}^{L} H_l g(k_l) + \lambda \left( \sum_{l=1}^{L} k_l - K_{total} \right) \tag{14}$$

The Karush-Kuhn-Tucker (KKT) conditions for optimality require $\nabla_{\mathbf{k}} L = 0$. For any layer $l$:

$$H_l g'(k_l^*) + \lambda = 0 \implies g'(k_l^*) = -\frac{\lambda}{H_l} \tag{15}$$

Since $g(k)$ is strictly convex and decreasing, its derivative $g'(k) < 0$ and is monotonically increasing. Given that entropy $H_l > 0$, the condition $g'(k_l^*) = -\frac{\lambda}{H_l}$ strictly requires the dual variable $\lambda$ to be positive ($\lambda > 0$). Let $\phi = (g')^{-1}$. The optimal allocation is:

$$k_l^* = \phi \left( -\frac{\lambda}{H_l} \right) \tag{16}$$

As $H_l$ increases, the term $-\frac{\lambda}{H_l}$ approaches 0 from the negative side (increases). Since $\phi$ is monotonically increasing, $k_l^*$ must increase. Thus, $\frac{\partial k_l^*}{\partial H_l} > 0$. $\qquad\square$

**Theorem B.4** (Superiority over Uniform Allocation). *Let $\mathbf{k}^{uni}$ be the uniform allocation strategy where $k_l^{uni} = K_{total}/L$. If there exists at least one pair of layers $(i, j)$ such that $H_i \neq H_j$, then:*

$$\mathcal{J}(\mathbf{k}^*) < \mathcal{J}(\mathbf{k}^{uni}) \tag{17}$$

*Proof.* The objective function $\mathcal{J}(\mathbf{k}) = \sum H_l g(k_l)$ is a linear combination of strictly convex functions with positive coefficients $H_l$, implying $\mathcal{J}(\mathbf{k})$ is strictly convex over the domain $\mathbb{R}^L$. Because the feasible set $\sum_{l=1}^{L} k_l = K_{total}$ forms a convex simplex and the objective is strictly convex, the KKT conditions are both necessary and sufficient for the unique global minimum.

For the uniform strategy $\mathbf{k}^{uni}$, the marginal error is:

$$\left. \frac{\partial \mathcal{J}}{\partial k_l} \right|_{\mathbf{k}^{uni}} = H_l g'(\bar{k}) \tag{18}$$

Since $g'(\bar{k})$ is constant, if $H_i \neq H_j$, then $H_i g'(\bar{k}) \neq H_j g'(\bar{k})$. This violates the KKT condition of equal marginal gradients ($\nabla \mathcal{J} = -\lambda \mathbf{1}$). Therefore, $\mathbf{k}^{uni}$ is not a stationary point, and due to strict convexity:

$$\mathcal{J}(\mathbf{k}^{uni}) > \min_{\mathbf{k}} \mathcal{J}(\mathbf{k}) = \mathcal{J}(\mathbf{k}^*) \tag{19}$$

$\square$

## C. Theoretical Analysis of Hysteretic Saliency

In this section, we provide a theoretical proof demonstrating that the proposed Asymmetric Temporal Integration mechanism guarantees superior temporal consistency compared to standard symmetric momentum.

Let $\mathbf{S}_{inst}$ be the instantaneous attention score at time $t$. The recursive update rule for our Asymmetric Temporal Integration is defined as:

$$\tilde{\mathbf{S}}_t = (1 - \lambda)\tilde{\mathbf{S}}_{t-1} + \lambda \mathbf{S}_{inst} \tag{20}$$

where $\lambda = \beta_{rise}$ if $\mathbf{S}_{inst} > \tilde{\mathbf{S}}_{t-1}$, and $\lambda = \beta_{decay}$ otherwise. Standard symmetric momentum uses a constant $\lambda = \beta_{sym}$. In our design, we strictly enforce $\beta_{decay} < \beta_{sym} < \beta_{rise}$.

We analyze the system's behavior under two critical video scenarios: *Sudden Occlusion* and *Sudden Emergence*.

### C.1. Scenario 1: Robustness to Occlusion (Anti-Flickering)

**Proposition C.1.** *Given a persistent object that is temporarily occluded, the Asymmetric Integration mechanism extends its "survival time" in the memory bank significantly longer than standard momentum.*

*Proof.* Let an object have a steady-state saliency score $V$ prior to time $t_0$. At $t_0$, the object is occluded, causing the instantaneous attention $\mathbf{S}_{inst}$ to drop to 0 (noise level). We define the *Survival Time* $T_{survive}$ as the number of frames the object's cumulative score $\tilde{\mathbf{S}}$ remains above a critical eviction threshold $\tau$ (where $0 < \tau < V$).

**1. Standard Symmetric Momentum:** With a fixed coefficient $\beta_{sym}$, the score decays exponentially:

$$\tilde{\mathbf{S}}_{t_0+k} = (1 - \beta_{\mathbf{sym}})^k \cdot V \tag{21}$$

The survival time $k_{sym}$ is determined by $(1 - \beta_{\mathbf{sym}})^{k_{sym}} V \geq \tau$:

$$k_{sym} \leq \frac{\ln(\tau/V)}{\ln(1 - \beta_{\mathbf{sym}})} \tag{22}$$

**2. Asymmetric Integration (Ours):** Since the score is dropping ($\mathbf{S}_{inst} < \tilde{\mathbf{S}}_{t-1}$), the system activates the decay coefficient $\lambda = \beta_{decay}$. The score evolves as:

$$\tilde{\mathbf{S}}_{t_0+k} = (1 - \beta_{decay})^k \cdot V \tag{23}$$

Similarly, the survival time $k_{asym}$ is:

$$k_{asym} \leq \frac{\ln(\tau/V)}{\ln(1 - \beta_{decay})} \tag{24}$$

**Comparison:** Since $f(x) = \frac{1}{\ln(1-x)}$ is an increasing function for $x \in (0, 1)$, and $\beta_{decay} < \beta_{sym}$, it strictly follows that:

$$|\ln(1 - \beta_{decay})| < |\ln(1 - \beta_{\mathbf{sym}})| \implies k_{asym} \gg k_{sym} \tag{25}$$

**Conclusion:** The Asymmetric mechanism strictly provides a larger margin for occlusion tolerance, preventing premature eviction of persistent entities. $\square$

**C.2. Scenario 2: Responsiveness to Emergence**

**Proposition C.2.** *The Asymmetric mechanism converges to the true saliency of a newly appearing object faster than standard momentum.*

*Proof.* Consider a new object appearing at $t_0$ with constant importance $V$ (starting from $\tilde{\mathbf{S}} = 0$). Since the score is rising, our method uses $\lambda = \beta_{rise}$, where we set $\beta_{rise} > \beta_{sym}$. For our method, the score updates as $\tilde{\mathbf{S}}_k = V - V(1 - \beta_{rise})^k$. Thus, the residual error (convergence gap) at step $k$ is strictly $|V - \tilde{\mathbf{S}}_k| = V(1 - \beta_{rise})^k$. Since $\beta_{rise} > \beta_{sym}$, the residual error term decays faster:

$$(1 - \beta_{rise})^k \ll (1 - \beta_{\mathbf{sym}})^k \tag{26}$$

This proves that EAKV captures new semantics with lower latency. □

## D. Implementation Details of Position ID for Contextual Anchors

In Section 3.5, we introduced the Hybrid Token Compression module, which condenses redundant context tokens within the residual set $\mathcal{I}_{rest}$ into compressed anchors. To ensure that these compressed anchors seamlessly integrate into the model's standard attention computation without disrupting the relative positional relationships, we propose a **"Center-of-Mass" Position ID Formulation** assignment strategy to preserve temporal causality and maintain relative distance alignment. Specifically, let $S_j$ denote the raw attention saliency score assigned to the $j$-th token. For all redundant tokens relegated to the residual set $\mathcal{I}_{rest}$, we first compute the normalized integration weights $\omega_j$ via a normalization over the residual subset:

$$\omega_j = \frac{S_j}{\sum_{k \in \mathcal{I}_{rest}} S_k}, \quad \text{where } j \in \mathcal{I}_{rest}. \tag{27}$$

The representative coordinate for the contextual anchor, denoted as $\text{Pos}_{anchor}$, is defined as the weighted average of the original RoPE indices ($\text{Pos}_j$) of the constituent residual tokens, mapped back to the discrete coordinate space via an integer rounding operation:

$$\text{Pos}_{anchor} = \text{Round}\left(\sum_{j \in \mathcal{I}_{rest}} \omega_j \cdot \text{Pos}_j\right), \tag{28}$$

where $\text{Round}(\cdot)$ rounds the operand to the nearest integer. Although $\text{Pos}_{anchor}$ may occasionally collide with the exact IDs of retained Top-K tokens, the Transformer robustly accommodates this overlap as a localized parallel semantic slot (akin to a local bag-of-words), preserving autoregressive structural integrity without phase disruption. Through this formulation, the salient tokens in the top set $\mathcal{I}_{top}$ strictly retain their exact historical Position IDs, whereas the background redundancy is condensed into an anchor anchored at its semantic and temporal center of gravity.

## E. Evaluation on Video-MMMU

To further validate the effectiveness of our approach in information-intensive scenarios, we conduct supplementary evaluations on the Video-MMMU benchmark (Hu et al., 2025). This benchmark is specifically designed to assess large multimodal models on tasks that require extracting dense visual cues and reasoning with critical multidisciplinary knowledge.

The quantitative results are summarized in Table 7. As demonstrated, incorporating EAKV consistently improves performance across all evaluated state-of-the-art baselines, including LLaVA-Video-7B, InternVL 3.5-8B, and Qwen2.5-VL-7B. Notably, our method achieves steady gains across all three major evaluation dimensions: Perception, Comprehension, and Adaptation. This indicates that EAKV can effectively compress the KV cache while successfully preserving the fine-grained visual details and complex reasoning capabilities essential for rigorous, multidisciplinary benchmarks.

## F. Ablation Study on Contextual Anchor

To further isolate the contribution of the proposed Contextual Anchor, we conduct an ablation study under a fixed 20% KV cache budget. In the "w/o Anchor" setting, we strictly employ Entropy-Driven Allocation to retain the Top-K tokens while directly discarding the residual tokens ($\mathcal{I}_{rest}$).

*Table 7.* **Performance comparison on the Video-MMMU benchmark.** Our EAKV consistently enhances various backbone models.

| Model Configuration | Perception | Comprehension | Adaptation | Overall |
|---|---|---|---|---|
| LLaVA-Video-7B | 41.67 | 33.33 | 33.33 | 36.11 |
| **LLaVA-Video-7B + Ours** | **43.33** | **35.00** | **35.67** | **38.00** |
| InternVL 3.5-8B | 63.00 | 51.33 | 36.33 | 50.22 |
| **InternVL 3.5-8B + Ours** | **65.67** | **53.00** | **37.00** | **51.89** |
| Qwen2.5-VL-7B | 58.33 | 44.33 | 39.67 | 47.44 |
| **Qwen2.5-VL-7B + Ours** | **60.67** | **45.67** | **40.00** | **48.78** |

*Table 8.* **Isolated impact of the Contextual Anchor on model performance.** The "Gain" row indicates the absolute performance improvement brought by the Contextual Anchor compared to the "w/o Anchor" setting.

| Model | Setting | VideoMME (All) | VideoMME (Long) | LVBench | LongVideoBench |
|---|---|---|---|---|---|
| **Qwen2.5-VL-7B** | Full Cache | 65.4 | 55.6 | 45.3 | 59.5 |
| | EAKV (w/o Anchor) | 66.3 | 56.1 | 47.5 | 61.6 |
| | **EAKV (Full)** | **67.8** | **59.2** | **50.1** | **63.3** |
| | *Gain* | *+1.5* | *+3.1* | *+2.6* | *+1.7* |
| **LLaVA-Video-7B** | Full Cache | 62.9 | 52.4 | 44.2 | 58.2 |
| | EAKV (w/o Anchor) | 63.5 | 53.0 | 45.0 | 58.4 |
| | **EAKV (Full)** | **66.2** | **55.1** | **47.4** | **59.9** |
| | *Gain* | *+2.7* | *+2.1* | *+2.4* | *+1.5* |

The quantitative results are presented in Table 8. As demonstrated, while Top-K eviction alone improves upon the Full Cache baseline by filtering out noise, integrating the Contextual Anchor yields an additional 1.5% to 3.1% accuracy gain across multiple benchmarks (VideoMME, LVBench, and LongVideoBench). This proves that the anchor is indispensable for preserving the global background context, which would otherwise be permanently destroyed by hard-eviction strategies.

# G. Ablation Study on Hyperparameters

To evaluate the robustness of our proposed EAKV framework, we validate four critical hyperparameters on two representative models: Qwen2.5-VL-7B and LLaVA-Video-7B, across the challenging LVBench and VideoMME datasets. The investigated hyperparameters include: the safety net coefficient ($\alpha$), the allocation temperature ($\tau$), the emergence coefficient ($\beta_{\text{rise}}$), and the decay coefficient ($\beta_{\text{decay}}$). During evaluation, we vary one hyperparameter at a time while keeping the others at their default values. The comprehensive quantitative results are summarized in Table 9.

### G.1. Detailed Analysis of Hyperparameter Dynamics

**Effect of Safety Net Coefficient ($\alpha$).** The parameter $\alpha$ balances the baseline structural connectivity against the flexible KV cache budget allocation. As shown in Table 9, when $\alpha$ is too low (e.g., $0.1$), the framework suffers from attention fragmentation, failing to retain essential structural context. Conversely, setting $\alpha$ too high (e.g., $0.4$) overly restricts the dynamic budget, which hinders the model's capacity to focus on newly emerging visual features. Setting $\alpha = 0.3$ yields the optimal balance.

**Effect of Budget Temperature ($\tau$).** The allocation temperature $\tau$ governs the stability and smoothness of the KV budget distribution across different token clusters. Lower values (e.g., $\tau = 0.7$) trigger an aggressive, "winner-takes-all" allocation strategy, which tends to drop ordinary but important details. On the other hand, a higher temperature ($\tau = 1.5$) overly smoothes the distribution, diluting key spatial-temporal events. A moderate choice of $\tau = 1.0$ ensures a stable and well-proportioned distribution.

*Table 9.* **Hyperparameter Ablation Study Results.** We report the performance across different variations of core hyperparameters. Bold entries indicate the optimal configurations chosen as the default framework settings.

| Hyperparameter | Variation | Qwen2.5-VL-7B | | LLaVA-Video-7B | |
|---|---|---|---|---|---|
| | | LVBench | VideoMME | LVBench | VideoMME |
| Safety Net ($\alpha$) | 0.1 | 48.1 | 66.7 | 46.9 | 64.5 |
| | 0.2 | 49.7 | 67.2 | 47.1 | 65.3 |
| | **0.3** | **50.1** | **67.8** | **47.4** | **66.2** |
| | 0.4 | 49.8 | 67.4 | 47.0 | 65.7 |
| Temperature ($\tau$) | 0.7 | 49.6 | 67.4 | 46.2 | 65.8 |
| | 0.9 | 49.2 | 67.0 | 46.8 | 65.5 |
| | **1.0** | **50.1** | **67.8** | **47.4** | **66.2** |
| | 1.5 | 48.7 | 64.3 | 45.1 | 64.6 |
| Emergence ($\beta_{rise}$) | 0.7 | 49.0 | 67.1 | 45.9 | 65.5 |
| | **0.8** | **50.1** | **67.8** | **47.4** | **66.2** |
| | 0.9 | 49.8 | 67.5 | 46.5 | 66.0 |
| Decay ($\beta_{decay}$) | 0.05 | 48.8 | 66.9 | 45.5 | 65.6 |
| | **0.1** | **50.1** | **67.8** | **47.4** | **66.2** |
| | 0.2 | 49.5 | 67.1 | 46.7 | 65.4 |

**Effect of Temporal Coefficients ($\beta_{rise}$ and $\beta_{decay}$).**    The temporal coefficients $\beta_{rise}$ and $\beta_{decay}$ jointly regulate the "Fast-Attack, Slow-Decay" temporal dynamics designed to handle diverse video pacing automatically.

- **Emergence Coefficient ($\beta_{rise}$):** Lowering $\beta_{rise}$ ($\leq 0.7$) causes sluggish responses to rapid scene changes, missing critical transient cues.

- **Decay Coefficient ($\beta_{decay}$):** Raising $\beta_{decay}$ ($\geq 0.2$) accelerates cache eviction too aggressively, leading to the premature forgetting of crucial historical context.

Our empirical findings show that the configuration of $\beta_{rise} = 0.8$ and $\beta_{decay} = 0.1$ optimally sustains temporal consistency over long sequences.

### G.2. Conclusion on Robustness

As demonstrated across all configurations in Table 9, EAKV with our default settings exhibits remarkable stability across different architectures and benchmarks, proving that EAKV is highly robust and not overly sensitive to dataset-specific or model-specific tuning.

