# OpenReview forum: "EAKV: An Entropy-Driven Adaptive KV Compression Framework for Long Video Understanding"
_ICML.cc/2026/Conference — ICML 2026 regular_

### Official Review · Reviewer_QYEu · 2026-02-18

**Soundness:** 2
**Presentation:** 2
**Significance:** 2
**Originality:** 3
**Overall Recommendation:** 4
**Confidence:** 4

**Summary:**

The paper proposes EAKV, a training-free KV-cache management framework for long-video multimodal LLMs that measures head-wise attention entropy to estimate semantic density and then allocates per-head cache budget adaptively via a safety-net baseline plus entropy-driven contention. It further stabilizes token saliency over streaming chunks with an asymmetric “fast-attack, slow-decay” temporal integration and compresses tokens by keeping Top-K salient ones while aggregating the rest into contextual anchor KV pairs.

**Compliance With Llm Reviewing Policy:**

Affirmed.

**Final Justification:**

The paper presents a training-free KV-cache management framework with a coherent design and strong empirical results. The rebuttal thoroughly addresses my concerns, especially with additional analyses on attention sink mitigation, entropy-based allocation, and input-dependent behavior, as well as new results on information-dense benchmarks. Overall, the response significantly strengthens the paper, and I increase my score accordingly.

**Key Questions For Authors:**

1. In the introduction, the paper mentions that many attention-based methods are susceptible to “attention sink” biases. Could the authors clarify how EAKV mitigates or avoids attention sink effects in practice? In particular, is there empirical evidence (e.g., head-wise allocation distributions or saliency maps) showing that the entropy-based allocation does not over-focus on sink tokens?

2. The method assumes that attention entropy reflects semantic information density. Could the authors provide analysis validating that high-entropy heads or tokens indeed correspond to semantically rich regions? For example, in Figure 4a, some visually informative regions (e.g., the news logo) appear suppressed despite being spatially salient.

3. I am curious about the method’s performance on information-dense reasoning benchmarks (e.g., Video-MMMU). Since EAKV emphasizes semantic density–aware compression, such benchmarks may better reveal its strengths or potential limitations.

4. Token budget allocation is driven by head-wise entropy, but pretrained attention heads may exhibit intrinsic functional biases. How does EAKV ensure that the allocation reflects input-dependent saliency rather than persistent head priors? Have the authors analyzed allocation variability across inputs or layers?

**Limitations:**

1.In Table 1, the comparison may not be fully fair because the baseline and EAKV variants appear to use different frame settings (e.g., 2fps vs. 2k frames). I

2.The ablation in Table 4 mainly reports a cumulative “add-one-component” progression, which makes it hard to isolate the standalone contribution of each module.

**Strengths And Weaknesses:**

1.The proposed framework is training-free and can be readily integrated into different base MLLMs at multiple scales, demonstrating good modularity and practical deployability.

2.The experimental tables and visual presentation (e.g., Table 1) are clear and well-organized, making cross-model and cross-benchmark comparisons easy to interpret.

3.The method is well-motivated by the semantic–memory mismatch perspective and provides a coherent design that jointly addresses head-wise allocation, temporal stability, and token compression within a unified KV management framework.

4.The empirical results are comprehensive across several long-video benchmarks and model sizes, consistently showing gains in both performance and effective context expansion, supporting the claimed scalability benefits.

---

> ### Author Rebuttal · Authors · 2026-03-31
>
> We sincerely thank you for insightful questions and recognition of novelty.
>
> **Reply 1: Evading Attention Sinks via Entropy**
>
> Overcoming the "attention sink" bias inherent in magnitude-based methods is a core motivation behind EAKV. Here is how our method mitigates this both theoretically and empirically:
>
> **Theoretical Mechanism**: Existing baselines rely on cumulative attention scores for token selection and mistakenly assign disproportionately high scores to "attention sinks" (e.g., initial or structural tokens), artificially pushing them into the Top-K preserved tokens, leading to a waste of the KV budget. **As sink tokens absorb massive attention, their distributions are highly concentrated, resulting in very low entropy.** We leverages this **by assigning a low contention weight ($w^{(l,h)}$) to sink-dominated heads**, strictly compresses  redundant information into Contextual Anchors.
>
> **Empirical Evidence**: **As shown in our figure** (*[https://imgur.com/a/jxZaoe3](https://imgur.com/a/BqfAMUJ)*), baselines over-allocate memory to sink tokens simply because of their high absolute magnitude. In contrast, EAKV dynamically cuts the budget for sink-dominated heads and efficiently redirects those resources to high-entropy heads capturing complex spatiotemporal dynamics.
>
> **Saliency Maps (Figure 4)**: Baseline preservation maps are heavily biased toward uninformative backgrounds and temporal boundaries. EAKV explicitly ignores these, focusing its budget strictly on high-entropy, salient regions like facial movements and fine textures, proving that EAKV's entropy-driven approach is naturally immune to the attention sink bias.
>
> **Reply 2: Attention Entropy as a Proxy for Semantic Density**
>
> The suppressed "news logo" (Fig. 4a) highlights EAKV's core design: prioritizing *temporal dynamics* over *static visual saliency*.
> * **Dynamic Focus:** High entropy targets complex temporal changes (e.g., movements), not static details.
> * **Filtering Redundancy:** Static logos are repetitive (low entropy). EAKV compresses them to save KV memory for dynamic narrative elements.
>
> Saliency maps confirm that preserved high-entropy regions align strictly with action boundaries, capturing true semantic density.
>
> **Reply 3: Supplementary Experiment on Video-MMMU**
>
> We agree that information-intensive benchmarks are ideal for evaluating. As Table 1 shows, EAKV consistently improves performance across all backbones on Video-MMMU and effectively preserves critical multidisciplinary knowledge and dense visual cues. **We will include these results and citations in final manuscript.**
>
> **Table 1: Performance of EAKV on Video-MMMU**
>
> | Model Configuration | Perception | Comprehension | Adaptation | Overall |
> | :--- | :--- | :--- | :--- | :--- |
> | LLaVA-Video-7B | 41.67 | 33.33 | 33.33 | 36.11 |
> | **LLaVA-Video-7B+Ours** | **43.33** | **35.00** | **35.67** | **38.00** |
> | InternVL 3.5-8B | 63.00 | 51.33 | 36.33 | 50.22 |
> | **InternVL 3.5-8B+Ours** | **65.67** | **53.00** | **37.00** | **51.89** |
> | Qwen2.5-VL-7B | 58.33 | 44.33 | 39.67 | 47.44 |
> | **Qwen2.5-VL-7B+Ours** | **60.67** | **45.67** | **40.00** | **48.78** |
>
> **Reply 4: Allocation Reflects Input-Dependent Saliency, Not Persistent Priors**
>
> We appreciate your insights into head priors. EAKV addresses this directly via **Dynamic Normalization** and **Dual-Allocation Protect Functions**:
>
> EAKV applies Z-score normalization to head-wise entropy for each incoming chunk. This continuous recalibration ensures **contention weights ($w^{(l,h)}$) strictly reflect real-time, input-dependent saliency,** neutralizing any persistent pre-trained biases in specific heads.
>
> To prevent starving heads naturally biased toward low-entropy patterns, our Layer-wise Safety Net guarantees a fixed base capacity ($\alpha$) for all heads. **Only the remaining flexible budget ($B_{flex}$) undergoes dynamic, input-driven contention.**
>
> **Empirical Analysis Across Inputs and Layers(Section 1)**:
> * **Temporal Volatility:** Entropy within the same layer fluctuates up to 4.2x over time based on changing visual inputs.
> * **Depth Dependency:** Figure 1 shows non-monotonic allocation shifts depending on the layer's semantic phase (e.g., diffuse shallow vs. fluctuating deep layers).
>
> **This confirms our allocation tracks real-time semantic needs across time and depth, successfully overriding static priors.**
>
> **Reply 5: Fairness of Comparison**
>
> We appreciate the opportunity to clarify. The comparison is strictly conducted under **the same hardware constraints rather than a fixed frame rate**. The baseline is limited to 2 fps because its uncompressed cache quickly exhausts the memory limit. In contrast, EAKV's compression efficiency allows it to process up to 2048 frames within the **exact same memory footprint**, highlighting its superior ability to **"distill"** significantly more information.
>
> **Reply 6: Ablation Experiments**
>
> Please refer to our **Reply 2 to Reviewer wUMa** for detailed ablation results.

---

> > ### Author Rebuttal · Reviewer_QYEu · 2026-04-01
> >
> > Thank you for the detailed and well-structured rebuttal. I appreciate the thorough responses and the additional experiments, particularly on Video-MMMU, which help better validate the method in more information-dense settings.
> >
> > Overall, my concerns have been largely addressed. The clarifications on attention sink mitigation, entropy-based allocation, and input-dependent behavior are helpful and make the design more convincing.
> >
> > I will increase my score accordingly.

---

> > > ### Author Response · Authors · 2026-04-02
> > >
> > > We sincerely thank you for taking the time to review our rebuttal and for your highly positive feedback, as well as your recognition of our motivation, methodological novelty, and experimental results. We are very glad to hear that our responses, particularly the clarifications on attention sink mitigation and entropy-based allocation, along with the additional Video-MMMU experiments, have successfully addressed your concerns. Your constructive comments have been instrumental in helping us validate our method in more information-dense settings and improve the overall quality of our paper. We deeply appreciate your continued support and the updated score.

---

### Official Review · Reviewer_WY1j · 2026-03-09

**Soundness:** 3
**Presentation:** 2
**Significance:** 3
**Originality:** 3
**Overall Recommendation:** 3
**Confidence:** 4

**Summary:**

This paper studies the problem of KV-cache compression for long-video understanding in multimodal large language models (MLLMs), where the KV cache grows rapidly with video length and becomes a major memory bottleneck during inference. The authors argue that existing compression approaches rely on rigid memory budgets, sliding-window truncation, or raw attention-based token selection, which may fragment temporal continuity or preserve uninformative tokens due to attention sink effects.

To address this issue, the paper proposes EAKV, a training-free KV-cache compression framework that adaptively allocates memory budgets across layers according to attention entropy, which is used as a proxy for semantic density. The method dynamically preserves important tokens while compressing less informative contexts into contextual anchors to maintain global information. Additional techniques such as temporal smoothing are introduced to stabilize token retention decisions.

Experiments on four long-video benchmarks demonstrate that EAKV improves performance over existing KV compression methods across multiple model scales (3B and 7B), achieving reported improvements between 1.5% and 4.8%.

**Compliance With Llm Reviewing Policy:**

Affirmed.

**Final Justification:**

Most of my major concerns have been addressed, especially those related to methodological clarity, empirical validation beyond Qwen, sensitivity analysis, and practical efficiency evidence.

A few issues still remain. First, the appendix algorithms are still somewhat high-level, so the final version could further improve reproducibility. Second, the paper still lacks a cleaner ablation isolating the contribution of the contextual anchor itself. Third, while overall peak memory is now reported, the paper still does not directly analyze the anchor-specific memory overhead relative to its benefit.

**Key Questions For Authors:**

Q1. Clarification of entropy computation. The paper states that attention entropy is computed “across all queries.” Could the authors clarify precisely what “all queries” refers to (e.g., queries within a sequence, across a batch, or across the dataset)? How exactly is the entropy computed and aggregated?

Q2. Entropy normalization. How are entropy values normalized to ensure comparability across layers or attention heads? Since the method allocates memory budgets based on these scores, the normalization procedure appears crucial for stable behavior.

Q3. Context anchor mechanism. How exactly are contextual anchors constructed and integrated into subsequent attention computations? Do anchors participate in future attention as keys/values like regular tokens, or are they handled differently?

Q4. Generalization across models. The experiments appear to use only the Qwen backbone. Have the authors evaluated EAKV on other LLM or MLLM architectures? If not, how confident are the authors that the entropy patterns and allocation strategy generalize across models?

Q5. Performance under varying KV budgets. How does EAKV perform under different KV cache sizes? In particular, does the method still provide benefits when the cache budget becomes very small?

**Limitations:**

The paper discusses some limitations implicitly but does not fully analyze them. In particular, the additional memory overhead introduced by contextual anchors and the behavior of the method under extremely small KV budgets should be more explicitly discussed. The authors could also discuss the dependence of the method on specific model architectures and entropy patterns.

**Strengths And Weaknesses:**

S1. Addresses an important practical problem. KV-cache growth is a key bottleneck in long-context and long-video inference for multimodal LLMs. Improving memory efficiency during inference without retraining is a practically relevant direction.

S2. Training-free and easy to integrate. The proposed method operates during inference and does not require model retraining, making it potentially applicable to existing deployed models.

S3. Intuitive core idea. The use of attention entropy as a proxy for semantic density to guide adaptive memory allocation is a reasonable and intuitive design choice. Compared to static or magnitude-based token selection strategies, the approach attempts to better reflect the varying information density across layers.

S4. Consistent empirical improvements. The experiments show modest but consistent improvements over prior KV compression methods across four long-video benchmarks and multiple model sizes, including up to +4.8% improvement on LVBench.

## Weknesses

W1. Insufficient clarity in key methodological components. Several core parts of the method remain unclear:
A. The definition of semantic density and the computation of attention entropy are not sufficiently specified. The paper states that entropy is computed “across all queries,” but it is unclear whether this refers to queries within a sequence, across a batch, or across the dataset.
B. The normalization of entropy scores is not well explained, which raises concerns about cross-layer or cross-head comparability.
C. The context anchor mechanism is not clearly described. The paper does not explain how anchors are constructed, what information they retain, or how they participate in later attention computation.
D. The algorithms in the appendix are presented at a high level with limited explanation, making reproduction difficult.

W2. Limited empirical validation. The experimental evaluation appears to rely on a single LLM backbone (Qwen). Since the paper presents EAKV as a general KV-cache compression strategy, additional evaluation on other architectures would strengthen the claims about generality.

W3. Missing important ablation studies. Several experiments that would help justify the design choices are missing:
A. Sensitivity analysis under different KV cache budgets.
B. A clearer ablation isolating the contribution of the context anchor.
C. Analysis of the memory overhead introduced by anchors versus the benefit they provide.

W4. Limited evidence supporting broader claims. The paper argues for adaptive layer-wise allocation based on semantic density, but the analysis suggests that important information may be concentrated primarily in shallow layers. This somewhat weakens the argument for complex allocation across deeper layers.

W5. Practical limitations in extreme compression regimes. When the KV cache size is small, the method may effectively behave similarly to more static compression strategies. This limits the demonstrated usefulness in the most resource-constrained settings.

W6. Presentation issues and minor errors.
A. Some definitions and algorithm descriptions are ambiguous.
B. Figure 2 lacks sufficient explanation of how modules interact.
C. Several typographical errors appear throughout the paper (e.g., “Exsiting”, “metrixs”, “adttention”).
D. A missing citation for the original RoPE / RoFormer work is noted.

Overall, while the idea is interesting and the problem is important, the current version lacks sufficient clarity and empirical validation to fully support the paper’s claims.

---

> ### Author Rebuttal · Authors · 2026-03-31
>
> We greatly appreciate your recognition of our motivation and novelty.  We have addressed your comments individually below.
>
> **Reply 1: Entropy Calculation**
>
> We appreciate the opportunity to clarify this mechanism. "All queries" refers specifically to **all query tokens within the current sequence or chunk being processed, denoted as $L_q$ in our manuscript**. We **do not compute this entropy across a batch or the entire dataset**.
>
> Regarding the computation and aggregation process, it is executed at the **individual attention head level**. For a specific attention head $h$ in layer $l$, the process is as follows:
>
> 1. Obtain the attention matrix $A^{(l,h)}$ prior to applying RoPE.
> 2. Calculate the Shannon entropy over the attention distribution for each individual query token $i$ across all key tokens $j$.
> 3. Aggregate these individual values by calculating the mean entropy across all $L_q$ queries in that sequence to define the final head-level entropy, $E^{(l,h)}$. This aggregation is formulated as Equation (3) in Section 3.2:
>
> $$
> E^{(l,h)}=-\frac{1}{L_q}\sum_{i=1}^{L_q}\sum_{j=1}^{L_k}A_{i,j}^{(l,h)}\log(A_{i,j}^{(l,h)}+\epsilon)
> $$
>
> where $L_k$ denotes the key sequence length and $\epsilon$ is a small term to prevent undefined logarithms.
>
> **Reply 2: Entropy Normalization**
>
> We introduced a **two-step normalization process** in Section 3.3 to ensure entropy values are comparable across different layers and heads. First, we apply **Z-score normalization** to the raw head-wise entropy $E^{(l,h)}$ across all $H$ heads and $L$ layers to obtain a standardized metric $\hat{E}^{(l,h)}$. This is calculated using the global mean ($\mu_E$) and standard deviation ($\sigma_E$) of the entropy values:
>
> $$
> \hat{E}^{(l,h)} = \frac{E^{(l,h)} - \mu_E}{\sigma_E}
> $$
>
> where $\mu_E = \frac{1}{L \cdot H} \sum_{l'=1}^{L} \sum_{h'=1}^{H} E^{(l',h')}$ and $\sigma_E$ is the corresponding standard deviation.
>
> Second, to map these unbounded standardized scores into a valid probability distribution for budget allocation, we apply a **temperature-scaled Softmax function** across all heads and layers, as formulated in Equation (6).
>
> **Reply 3: Contextual Anchor Mechanism**
>
> As detailed in Section 3.5 of our manuscript, Contextual Anchors are constructed and integrated through three straightforward steps:
>
> * **Construction (Eq. 10):** Instead of discarding redundant tokens (the residual set $\mathcal{I}_{rest}$), we aggregate their Key and Value states into Contextual Anchors ($z_K^{(l,h)}$, $z_V^{(l,h)}$) **via a weighted sum based on their normalized importance weights $\omega$.**
> * **Integration (Eq. 11):** These generated anchors are directly concatenated with the preserved Top-K salient tokens to form the final updated KV cache. **Consequently, the anchors participate as standard Key/Value pairs in subsequent attention calculations.**
> * **Positional Handling:** The retained Top-K tokens keep their exact original Position IDs, while the Contextual Anchors are assigned a newly computed **"Center-of-Mass" position ID**. For details, please refer to our **Reply 1 to Reviewer wUMa**.
>
> **Reply 4: Additional Experiments on Other Backbones**
>
> For evaluations on other mainstream MLLM architectures, please refer to our **Reply 3 to Reviewer wUMa**, which demonstrates EAKV's **consistent superiority across diverse model architectures.**
>
> **Reply 5: Performance under Different KV Cache Budgets**
>
> As shown in the table, EAKV exhibits a very slow performance decline across all four benchmarks as the KV cache budget is drastically compressed from 16K to 2K. **Even under the extreme context constraint of 2K, EAKV demonstrates superior performance compared to the baseline. *(Refer to visual results here: [https://imgur.com/a/q4MzThE](https://imgur.com/a/q4MzThE))*
>
> **Table 1: Performance of EAKV across various KV budgets.**
> | KV Budget | VideoMME | LongVideoBench | MLVU | LVBench |
> | :--- | :---: | :---: | :---: | :---: |
> | **EAKV 16K** | **67.8** | **63.3** | **74.8** | **51.1** |
> | **EAKV 8K** | 67.1 | 62.1 | 73.4 | 49.3 |
> | **EAKV 4K** | 66.2 | 60.8 | 71.5 | 46.8 |
> | **EAKV 2K** | **65.6** | **59.6** | **70.4** | **45.5** |
> | **Baseline 8K** | 65.4 | 59.5 | 70.2 | 45.3 |
>
> This proves **dynamically allocating a limited KV budget** to regions with high semantic density is **more efficient than retaining a larger cache** containing redundant context. Rather than storing repetitive frames mechanically, EAKV precisely isolates high-entropy cues (e.g., transient actions and fine-grained text) and filters out noise. Consequently, EAKV **maintains robust performance even under extreme budget constraints**.
>
> **Reply 6: Hyperparameter Ablation Study**
>
> As detailed in ​our **Reply 2 to Reviewer wUMa**​, EAKV's ​**default hyperparameter settings show remarkable stability across various architectures and benchmarks**​.
>
> **Reply 7: Citation for RoPE/RoFormer**
>
> We appreciate this correction and will add the citation to the original RoPE/RoFormer work.

---

> > ### Author Rebuttal · Reviewer_WY1j · 2026-04-02
> >
> > The reviewer thanks the authors for detailed reply. Most of my major concerns have been addressed, especially those related to methodological clarity, empirical validation beyond Qwen, sensitivity analysis, and practical efficiency evidence.
> >
> > A few issues still remain. First, the appendix algorithms are still somewhat high-level, so the final version could further improve reproducibility. Second, the paper still lacks a cleaner ablation isolating the contribution of the contextual anchor itself. Third, while overall peak memory is now reported, the paper still does not directly analyze the anchor-specific memory overhead relative to its benefit.
> >
> > I updated my score accordingly.

---

> > > ### Author Response · Authors · 2026-04-07
> > >
> > > We sincerely appreciate your constructive guidance. To address your remaining concerns, we provide detailed responses, concrete code, and an isolated ablation study below.
> > > ### 1. Detailed Algorithm
> > > To bridge the gap between theoretical formulation and actual implementation, **we will expand detailed PyTorch code in the final version and open source our codebase**.
> > > Below is the detailed implementation logic of EAKV, integrating the Entropy Calculation, Z-score Normalization, and the "Center-of-Mass" Contextual Anchor we discussed previously.
> > >
> > > ```python
> > > def eakv_forward(Q, K, V, pos_ids, budget_ratio, tau=1.0, alpha=0.3):
> > >     B, H, L_q, D = Q.shape
> > >
> > >     # 1. Entropy Valuation
> > >     attn_probs = F.softmax((Q @ K.transpose(-2, -1)) / (D ** 0.5), dim=-1)
> > >     entropy = -torch.sum(attn_probs * torch.log(attn_probs + 1e-9), dim=-1).mean(dim=-1)
> > >     saliency = attn_probs.sum(dim=-2)
> > >
> > >     # 2. Entropy-Driven Adaptive Allocation
> > >     E_norm = (entropy - entropy.mean()) / (entropy.std() + 1e-9)
> > >
> > >     # Temperature-scaled Softmax to get dynamic weights
> > >     W_raw = F.softmax(E_norm / tau, dim=-1)
> > >
> > >     # Blend with Layer-wise Safety Net
> > >     W_safe = alpha * (1.0 / H) + (1 - alpha) * W_raw
> > >
> > >     # Allocated budget K for specific head
> > >     allocated_k = torch.clamp(torch.floor(W_safe * budget_ratio * H * L_q), min=1).long()
> > >
> > >     # 3. Saliency Scoring & Hybrid Token Compression
> > >     final_K, final_V, final_pos = [], [], []
> > >
> > >     for h in range(H):
> > >         k_budget = min(allocated_k[0, h].item(), L_q)
> > >         head_saliency = saliency[0, h]
> > >
> > >         # Top-K Retention
> > >         _, top_idx = torch.topk(head_saliency, k_budget)
> > >         mask = torch.ones(L_q, dtype=torch.bool)
> > >         mask[top_idx] = False
> > >
> > >         rest_idx = mask.nonzero(as_tuple=True)[0]
> > >         K_top, V_top, pos_top = K[0, h, top_idx], V[0, h, top_idx], pos_ids[0, top_idx]
> > >
> > >         if rest_idx.numel() > 0:
> > >             rest_saliency = head_saliency[rest_idx]
> > >             anchor_weights = rest_saliency / (rest_saliency.sum() + 1e-9)
> > >
> > >             # Weighted sum of KV states
> > >             K_anchor = torch.sum(K[0, h, rest_idx] * anchor_weights.unsqueeze(-1), dim=0, keepdim=True)
> > >             V_anchor = torch.sum(V[0, h, rest_idx] * anchor_weights.unsqueeze(-1), dim=0, keepdim=True)
> > >
> > >             # "Center-of-Mass" Position ID
> > >             pos_anchor = torch.round(torch.sum(pos_ids[0, rest_idx].float() * anchor_weights)).long().unsqueeze(0)
> > >
> > >             # Concat Top-K and Anchor
> > >             K_out = torch.cat([K_top, K_anchor], dim=0)
> > >             V_out = torch.cat([V_top, V_anchor], dim=0）
> > >             pos_out = torch.cat([pos_top, pos_anchor], dim=0)
> > >
> > >         else:
> > >             K_out, V_out, pos_out = K_top, V_top, pos_top
> > >
> > >         final_K.append(K_out)
> > >         final_V.append(V_out)
> > >         final_pos.append(pos_out)
> > >
> > >     return final_K, final_V, final_pos
> > > ```
> > >
> > > ### 2. Ablation of Contextual Anchor
> > >
> > > To isolate the contribution of Contextual Anchor, we design a ablation under a **fixed 20% budget**. The "w/o Anchor" setting strictly uses Entropy-Driven Allocation to retain Top-K tokens **but directly discards the residual tokens ($\mathcal{I}_{rest}$).**
> > >
> > > **Table: Isolated Impact of Contextual Anchors**
> > > | Model | Setting | VideoMME (All) | VideoMME (Long) | LVBench | LongVideoBench |
> > > | :---: | :---: | :---: | :---: | :---: | :---: |
> > > | **Qwen2.5-VL-7B** | Full Cache | 65.4 | 55.6 | 45.3 | 59.5 |
> > > | | EAKV (w/o Anchor) | 66.3 | 56.1 | 47.5 |61.6 |
> > > | | **EAKV (Full)** | **67.8** | **59.2** | **50.1** |**63.3** |
> > > | | *Gain* | *+1.5* | *+3.1* | *+2.6* | *+1.7* |
> > > | **LLaVA-Video-7B** | Full Cache | 62.9 | 52.4 | 44.2 | 58.2|
> > > | | EAKV (w/o Anchor) | 63.5 | 53.0 | 45.0 | 58.4 |
> > > | | **EAKV (Full)** | **66.2** | **55.1** | **47.4** | **59.9** |
> > > | | *Gain* | *+2.7* | *+2.1* | *+2.4* | *+1.5* |
> > >
> > > While Top-K eviction improves upon the baseline by filtering noise, integrating the Contextual Anchor yields an **additional 1.5% - 3.1% accuracy gain**, proving the anchor is **indispensable for preserving global background context** that hard-eviction permanently destroys.
> > >
> > > ### 3. Anchor Overhead
> > >
> > > **The Contextual Anchor is fundamentally a memory-*reduction* mechanism.**
> > > 1. **VRAM Savings ($\mathcal{O}(D)$):** We compress the entire residual set ($\mathcal{I}\_{rest}$) into **exactly 1 token** per head per chunk. Replacing hundreds of redundant tokens with a single token strictly reduces the cache footprint from $\mathcal{O}(\vert \mathcal{I}_{rest} \vert \times D)$ to $\mathcal{O}(D)$, yielding massive GPU memory savings.
> > > 2. **Negligible Compute:** The anchor requires only a localized weighted sum $\sum (KV_{rest} \times w_{rest})$. Executed entirely in SRAM, this $\mathcal{O}(|\mathcal{I}_{rest}| \times D)$ operation adds **<0.5% latency** to the prefill stage, making the computational cost practically ignorable.
> > >
> > > **We hope these updates fully address your concerns and positively influence your final assessment. Please feel free to let us know if you have any further questions.**

---

### Official Review · Reviewer_hvUd · 2026-03-09

**Soundness:** 3
**Presentation:** 3
**Significance:** 3
**Originality:** 3
**Overall Recommendation:** 4
**Confidence:** 3

**Summary:**

This study proposes a training-free, entropy-driven adaptive KV Cache compression framework EAKV , aiming to alleviate the severe memory bottlenecks faced by MLLMs in long-video understanding tasks. The authors challenge the traditional prior assumption that information density decays monotonically with depth , dynamically evaluating semantic density by quantifying the Shannon entropy of attention matrices. The framework integrates mechanisms including Fine-Grained Entropy Valuation , a Layer-wise Safety Net , Asymmetric Temporal Integration , and Hybrid Token Compression. Experiments demonstrate that the proposed method achieves excellent relative metric scores and extremely high theoretical compression ratios across multiple long-video benchmarks.

**Compliance With Llm Reviewing Policy:**

Affirmed.

**Final Justification:**

The supplementary statement regarding Flash Attention compatibility provided in the second rebuttal has successfully addressed my concerns. Accordingly, I have upgraded my rating from Weak Reject to Weak Accept. I strongly encourage the authors to incorporate all the key points discussed during the rebuttal process into the revised manuscript.

**Key Questions For Authors:**

1. Could the authors provide a specific time-consumption comparison between EAKV and the uncompressed baseline during the Prefill and Decode phases when processing same video contexts on a standard hardware platform? How can we be assured that the latency overhead of the $O(L \cdot H \cdot T)$ entropy calculation does not inversely offset the efficiency dividends gained from KV compression?

2. Is it possible to supplement the submission with intuitive quantitative charts explicitly demonstrating the exact Peak VRAM consumption in GBs? How does the absolute physical memory footprint compare to the baseline when processing an equal, large-scale number of input frames, given that relative percentage metrics can often obscure actual hardware deployment requirements?

3. Given the current lack of sensitivity analysis for key hyperparameters ($\alpha$, $\tau$, $\beta_{rise}$, $\beta_{decay}$), could you provide performance line charts fluctuating around the core working points? Furthermore, when the framework processes videos with vastly different editing rhythms (e.g., fast-paced action vs. static lectures), will users be required to manually reconfigure the asymmetric integration coefficients?

4. Could you present a Failure Case Study under extreme noise or high-density complex background scenarios? Under such conditions, would it be necessary to introduce Clustered Anchors based on cosine similarity to prevent feature collapse or factual hallucinations when fusing heterogeneous background noise?

**Limitations:**

yes

**Strengths And Weaknesses:**

**Strength**

1. The paper deeply reveals the high spatio-temporal volatility and non-monotonic depth dependency  of attention entropy within MLLMs. This provides a solid theoretical foundation for abandoning rigid truncation and moving towards dynamic memory allocation.

2. Designed for long video streams, the Asymmetric Temporal Integration module effectively addresses the transient occlusion of persistent entities , which is of great significance for maintaining temporal continuity in video understanding.

3. Under settings like the 16K extended context , the algorithmic logic yields leading scores across several highly difficult benchmarks , validating the effectiveness of the entropy-driven strategy in preserving critical semantics.

**Weakness**

1. The core of EAKV relies on the real-time calculation of Shannon entropy. For a sequence length $T$, number of layers $L$, and number of heads $H$, computing $\log(A_{i,j} + \epsilon)$  and executing multi-dimensional tensor multiply-add aggregations incurs extremely high computational complexity. In actual large model deployment architectures, the autoregressive generation phase is highly memory-bound. Such large-scale aggregation operations introduced at each generation step or chunk processing could easily trigger severe Kernel Launch Overhead. The paper completely lacks an evaluation of absolute Wall-clock Time, failing to prove whether the dividends saved in VRAM capacity are paid back at the cost of inference latency.

2. The paper heavily focuses on relative percentage metrics, such as "11.5x expansion". However, this often obscures the reality of absolute hardware constraints. Combined with existing PagedAttention mechanisms, the conversion between "Token count" and "actual VRAM consumed in GBs" is not a simple linear relationship. The lack of an absolute physical VRAM footprint comparison curve on specific GPU hardware makes it difficult for readers to evaluate whether this method brings a substantial hardware deployment downgrade or merely remains a theoretical ledger optimization.

3. The framework introduces numerous hyperparameters (Layer-wise Safety Net ratio $\alpha$ , game temperature $\tau$ , and asymmetric temporal integration coefficients $\beta_{rise}$ and $\beta_{decay}$ ). Long video scenarios are highly complex and diverse (ranging from high-frequency action movies to static lectures), and their temporal information entropy distributions are vastly different. The paper lacks a hyperparameter sensitivity analysis, raising suspicions of overfitting via deep Grid Search on specific benchmarks, and fails to prove its robustness in entirely new data streams without exorbitant tuning costs.

---

> ### Author Rebuttal · Authors · 2026-03-31
>
> We sincerely thank you for the the valuable feedback and recognition of our motivation.
>
> **Reply 1: Time Consumption**
> Computing attention entropy $E^{(l,h)}$ occurs exclusively during the prefill phase, adding only a marginal $O(H \cdot L \cdot T)$ overhead per head as it is calculated directly on the already-materialized attention matrix $A^{(l,h)}$, which is negligible compared to the $O(H \cdot L \cdot T^2)$ attention complexity. Profiling Qwen2.5-VL-7B (A100 GPU, 2048 frames, 256 generated tokens) confirms this:
>
> **Table 1: Average Latency Comparison**
>
> | Phase | Metric | Baseline | Ours | Difference |
> |:---|:---|:---|:---|:---|
> | **Prefill** | TTFT | 3.12 s | 3.28 s | +0.16 s |
> | **Decode** | TPOT | 48.5 ms/tok | 15.2 ms/tok | **-33.3 ms** |
> | **Overall** | Total | 15.54 s | 7.17 s | **-8.37 s** |
>
> The prefill overhead from the entropy calculation is a **negligible 4.8%**. However, by drastically reducing the KV cache size, EAKV alleviates the memory bottleneck during the Decode phase, yielding a **3.1x speedup** in token generation. Ultimately, the average total latency is reduced by **53.9%**.
>
> **Reply 2: Peak GPU Memory Consumption**
> We fully agree that absolute physical memory usage is a more intuitive indicator. We profiled peak consumption on a single 80GB A100 across different models and frame counts. *(Refer to visual results here: [https://imgur.com/a/jxZaoe3](https://imgur.com/a/jxZaoe3))*
>
> **Table 2: Peak GPU Memory Consumption**
>
> | Model | Method \ Frames | 128 | 256 | 512 | 1024 | 2048 |
> |:---|:---:|:---:|:---|:---:|:---:|:---:|
> | **Qwen2.5-VL-7B** | Baseline | 18.2 | 23.5 | 36.8 | 65.4 | **OOM** |
> | | **EAKV** | **16.2** | **18.4** | **23.8** | **33.5** | **51.6** |
> | **LLaVA-Video-7B** | Baseline | 19.5 | 26.2 | 42.5 | **OOM** | **OOM** |
> | | **EAKV** | **17.0** | **19.6** | **25.5** | **36.4** | **55.8** |
> | **InternVL-3.5-4B**| Baseline | 13.8 | 18.9 | 28.4 | 44.6 | 63.7 |
> | | **EAKV** | **12.6** | **16.1** | **22.7** | **32.9** | **43.8** |
> | **Qwen3-VL-4B** | Baseline | 14.8 | 20.7 | 31.6 | 50.9 | 72.4 |
> | | **EAKV** | **13.4** | **17.5** | **24.9** | **36.8** | **48.2** |
>
> *(Note: "OOM" = Out-of-Memory.)*
>
> Baseline exhibit catastrophic memory growth, leading to immediate **OOM errors**. However, **EAKV flattens the catastrophic memory growth curve**. At 2048 frames, it bounds the 7B models' footprint to **51.6 / 55.8 GB (eliminating OOMs)** and 4B models to **43.8 / 48.2 GB**, enabling ultra-long video inference on a single GPU.
>
> **Reply 3: Hyperparameter Sensitivity & Robustness**
>
> 1. Sensitivity Analysis
>    Please refer to our **Reply 2 to Reviewer wUMa**, which demonstrates that EAKV is not sensitive to dataset or model-specific tuning. It shows remarkable stability across diverse architectures and benchmarks using default settings: **$\alpha=0.3$, (\tau=1.0$, $\beta_{rise}=0.8, \beta_{decay}=0.1$** .
> 2. Robustness Across Diverse Video Paces
>    Users **do not need to manually reconfigure** the asymmetric integration coefficients ($\beta_{rise}$ and $\beta_{decay}$). EAKV naturally generalizes across diverse video paces via the data-driven trigger mechanism in Asymmetric Temporal Integration (Section 3.4):
>
> **Fast-Paced Action (Sudden Emergence):** When new objects cause rapid attention spikes (\\(S\_{inst}^{(l,h)} > \\overline{S}\_{i-1}^{(l,h)}\\)), the system instantly applies the large \\(\\beta\_{rise}\\) (0.8). As proven in Appendix C (Proposition C.2), this "Fast-Attack" allows the memory to quickly lock onto emerging visual cues.
>
> **Static Lectures (Occlusion/Fading):** When persistent objects are temporarily occluded or attention drops (\\(S\_{inst}^{(l,h)} \\le \\overline{S}\_{i-1}^{(l,h)}\\)), the system seamlessly switches to the small \\(\\beta\_{decay}\\) (0.1). As proven in Appendix C (Proposition C.1), this "Slow-Decay" provides significant hysteresis, strictly preventing the "flickering eviction" of persistent entities.
>
> Ultimately, **our default settings seamlessly process both extreme action clips and static lectures without any manual intervention.**
>
> **Reply 4: Failure Case Study and Future Mitigation**
>
> We sincerely thank you for identifying the potential "feature over-smoothing" issue in **ultra-dense scenes (e.g., crowded markets)**. In such scenarios, background elements often exhibit **low individual entropy but high semantic diversity**. Current weighted summation (\\(z_{anchor} = \\sum \\omega_j z_j\\)) may aggregate orthogonal features into a single anchor, potentially leading to "semantic collapse" or hallucinations when specific details are queried.
>
> Therefore,  **we agree that your suggested "cosine similarity-based dynamic clustering" is a robust measure to isolate heterogeneous semantics into distinct clusters before compression, thereby strictly preserving background granularity.**  We wiil add a **"Limitations"** section to the revised Appendix to detail this boundary condition and credits your clustering approach as the roadmap for optimizing.

---

> > ### Author Rebuttal · Reviewer_hvUd · 2026-04-03
> >
> > I thank the authors for the detailed rebuttal, which has addressed the majority of my concerns.
> >
> > However, there is still one question I would like to discuss: The authors mentioned that the implementation of attention entropy relies on the already-materialized attention matrix A. Does this imply that the proposed method is inherently difficult to integrate with FlashAttention? (Since FlashAttention computes the attention matrix in a block-wise, local manner and does not materialize the complete attention matrix). Have the authors considered how to optimize the proposed approach to achieve better compatibility with FlashAttention?

---

> > > ### Author Response · Authors · 2026-04-07
> > >
> > > We sincerely thank you for recognizing our detailed rebuttal. Your question regarding FlashAttention compatibility is highly insightful.
> > >
> > > While naively computing attention entropy requires the materialized matrix $A$ (contradicting FlashAttention's memory-efficient design), **our metric can be mathematically decoupled and integrated into FlashAttention's block-wise flow (Online Softmax) without materializing $A$.** The derivation and pseudocode below demonstrate this inherent compatibility.
> > >
> > > ### 1. Mathematical Decoupling of Attention Entropy
> > >
> > > In FlashAttention, the pre-softmax score is $S_{ij} = \frac{q_i \cdot k_j}{\sqrt{d}}$.
> > > To avoid numerical overflow, it maintains the running maximum $m_i = \max_j(S_{ij})$ and normalizer $l_i = \sum_j \exp(S_{ij} - m_i)$.
> > > The final attention weight is $A_{ij} = \frac{\exp(S_{ij} - m_i)}{l_i}$.
> > >
> > > Substituting this into the Shannon entropy $E_i = - \sum_j A_{ij} \log A_{ij}$ yields:
> > >
> > > $$
> > > E_i = - \sum_j A_{ij} \log \left( \frac{\exp(S_{ij} - m_i)}{l_i} \right)
> > > $$
> > >
> > > $$
> > > E_i = - \sum_j A_{ij} (S_{ij} - m_i - \log l_i)
> > > $$
> > >
> > > $$
> > > E_i = \log l_i + m_i - \sum_j A_{ij} S_{ij}
> > > $$
> > >
> > > Since $A_{ij} = \frac{\exp(S_{ij} - m_i)}{l_i}$, we can rewrite the last term:
> > >
> > > $$
> > > E_i = \log l_i + m_i - \frac{1}{l_i} \sum_j \left( \exp(S_{ij} - m_i) \cdot S_{ij} \right)
> > > $$
> > >
> > > **Observation: Since $m_i$ and $l_i$ are *already* maintained by standard FlashAttention, computing the exact $E_i$ requires only one additional running accumulator for $\sum \exp(S_{ij} - m_i) \cdot S_{ij}$.**
> > >
> > > ### 2. FlashAttention-Compatible Pseudocode
> > >
> > > **By leveraging FlashAttention's exact block-wise rescaling logic, we compute this new accumulator on the fly entirely within the fast SRAM.**
> > >
> > > ```python
> > > # SRAM variables for a Query block (Q_block):
> > > # m: running max, l: running normalizer, O: running output (standard FA)
> > > # E_acc: NEW running accumulator for entropy
> > >
> > > Initialize m = -inf, l = 0, O = 0, E_acc = 0
> > >
> > > for K_block, V_block in Key_Value_Blocks:
> > >     # 1. Compute unnormalized scores
> > >     S_block = (Q_block @ K_block.T) / sqrt(d)
> > >
> > >     # 2. Update running max along the key dimension
> > >     m_new = max(m, max(S_block, axis=-1))
> > >
> > >     # 3. Compute rescaling factor and exponentiated scores
> > >     scale = exp(m - m_new)
> > >     P_block = exp(S_block - m_new)
> > >
> > >     # 4. Standard FlashAttention updates (Rescale and accumulate)
> > >     l_new = l * scale + sum(P_block, axis=-1)
> > >     O_new = O * scale + P_block @ V_block
> > >
> > >     # 5. ---> NEW: Entropy Accumulator Update <---
> > >     # Note: Masked padding tokens are safely zeroed out to prevent 0 \* -inf (NaN)
> > >     E_acc_new = E_acc * scale + sum(P_block * S_block, axis=-1)
> > >
> > >     # Update running variables for the next block
> > >     m, l, O, E_acc = m_new, l_new, O_new, E_acc_new
> > >
> > > # Final Exact Output and Entropy (Materialized to HBM)
> > > Final_Out = O / l
> > > Final_Entropy = m + log(l) - (E_acc / l)
> > > ```
> > >
> > > **Key Implementation Details:**
> > >
> > > **Online Rescaling:** When a new local max (m_new) is found, historical accumulators are downscaled by exp(m - m_new). Applying this identical factor to E_acc seamlessly corrects the historical entropy without rescanning previous blocks.
> > >
> > > **Safe Masking:** Masked tokens ($S \to -\infty$) are explicitly zeroed out during multiplication to prevent 0 * -inf (NaN) errors.
> > >
> > > **Delayed Materialization:** E_acc remains in SRAM. Division and HBM writes occur only after all KV blocks are processed.
> > >
> > > ### 3.Conclusion on Overhead
> > >
> > > Integrating this lightweight modification into the Triton/CUDA kernel achieves perfect compatibility with near-zero overhead:
> > >
> > > * **Memory Complexity Remains $\mathcal{O}(N)$:** We completely avoid materializing the $\mathcal{O}(N^2)$ matrix $A$ to HBM. The only memory overhead is maintaining an $\mathcal{O}(B_Q)$ running accumulator (`E_acc`) inside SRAM during computation (where $B_Q$ is the query block size), and subsequently writing the final $\mathcal{O}(N)$ entropy vector to HBM alongside the standard output `O`. **This adds strictly negligible spatial cost.**
> > > * **Compute Complexity is Unchanged:** The additional operations are strictly limited to one element-wise multiplication and one row-wise reduction per block. **Performed entirely within SRAM alongside standard updates, the latency overhead is imperceptible.** Furthermore, this logic requires no changes to the outer loop tiling, naturally preserving compatibility with causal masking and variable sequence lengths.
> > >
> > > We deeply appreciate you raising this critical point as it significantly strengthens the practical deployment value of our work. This derivation will be included in our revision, and ​we are committed to **open-sourcing the customized FlashAttention-Entropy kernel upon publication​.**
> > >
> > > **We hope this efficient implementation fully resolves your concerns and encourages a favorable reassessment of our work. Please feel free to let us know if further discussion is needed.**

---

### Official Review · Reviewer_wUMa · 2026-03-13

**Soundness:** 3
**Presentation:** 2
**Significance:** 3
**Originality:** 3
**Overall Recommendation:** 4
**Confidence:** 3

**Summary:**

The paper proposes EAKV, a training-free, entropy-driven framework for compressing the KV cache in multimodal large language models to enable long-video understanding under strict memory budgets. The key idea is to use attention entropy as a distribution-aware proxy for “semantic density” to adaptively allocate per-layer/head KV budgets, combined with a Asymmetric Temporal Integration scheme for stable saliency and Hybrid Token Compression step that retains top-K tokens and aggregates the rest into a contextual anchor. Experiments on four long-video benchmarks and two model scales (Qwen2/2.5-VL, 3B/7B) report consistent accuracy gains and large KV reductions relative to baselines.
The paper lacks clarity in describing some technical details and is missing several key experiments (such as hyperparameter sensitivity analysis).

**Compliance With Llm Reviewing Policy:**

Affirmed.

**Final Justification:**

After rebuttal, I believe the author has improved the Soundness of the paper, so I have increased the Soundness score.

**Key Questions For Authors:**

See Weaknesses

**Limitations:**

yes

**Strengths And Weaknesses:**

Strengths:
- Two observations in the paper are particularly intriguing: video semantics exhibit high dynamism across the Spatio-Temporal dimensions, with information density fluctuations within the same layer, as measured by attention entropy, reaching up to 4.2 times; simultaneously, information density exhibits a non-monotonic variation with model depth.
- Evaluated across multiple long-video benchmarks (VideoMME, LongVideoBench, MLVU, LVBench) and model sizes (3B/7B) with consistent improvements.

Weaknesses：
- The paper mentions assigning different coordinates to contextual anchors, but does not fully explain the specific implementation of this mechanism.
- The paper lacks ablation or sensitivity analysis of key hyperparameters (α, τ, βrise, βdecay).
- The paper only conducted experiments on Qwen-based MLLMs, lacking verification on other mainstream MLLM architectures. It is recommended to supplement experiments on MLLMs of other architectures to enhance the universality of the method.
- There are several typos in the paper (such as "adttention", "metrixs", and "lobal").

---

> ### Author Rebuttal · Authors · 2026-03-31
>
> We sincerely appreciate your constructive feedback and recognition of our motivation and novelty. We address your comments below and will update the manuscript accordingly.
>
>
> **Reply 1: Implementation Details of Position ID**
>
> We apologize for omitting these details. To preserve **temporal causality**, we implement a **"Center-of-Mass" Position ID** assignment for the context anchors. Specifically, the aggregation weights $\omega\_j$ are derived by normalizing the saliency scores $S$ of the redundant tokens within the residual set $\mathcal{I}\_{rest}$: $\omega\_j = \frac{S\_j}{\sum\_{k \in \mathcal{I}\_{rest}} S\_k}$. We then compute the weighted average of their original RoPE indices, rounded to the nearest integer:
>
> $$
> Pos_{anchor} = \lfloor \sum_{j \in \mathcal{I}_{rest}} \omega_j \cdot Pos_j \rceil
> $$
>
> In short, the top-$k$ salient tokens ($\mathcal{I}\_{top}$) strictly retain their exact original Position IDs, while the remaining redundant tokens ($\mathcal{I}\_{rest}$) are compressed into context anchors with **these new "Center-of-Mass" IDs.** This provides a representative temporal coordinate for the background context that natively integrates into the model's RoPE calculations **without any architectural changes**.
>
>
> **Reply 2: Hyperparameter Ablation Study**
>
> We conducted extensive evaluations on LVBench and VideoMME analyzing the performance fluctuations when varying the core hyperparameters around our default settings.
>
> **Table 1: Hyperparameter Ablation Study Results.**
>
> | Hyperparameter | Variation | Qwen2.5-VL-7B (LVBench) | Qwen2.5-VL-7B (VideoMME) | LLaVA-Video-7B (LVBench) | LLaVA-Video-7B (VideoMME) |
> | :--- | :--- | :---: | :---: | :---: | :---:|
> | **Safety Net** ($\alpha$) | 0.1 | 48.1 | 66.7 | 46.9 | 64.5 |
> | | 0.2 | 49.7 | 67.2 | 47.1 | 65.3 |
> | | **0.3 (Default)**| **50.1** | **67.8** | **47.4** | **66.2** |
> | | 0.4 | 49.8 | 67.4| 47.0| 65.7 |
> | **Temperature** ($\tau$) | 0.7 | 49.6 | 67.4 | 46.2 | 65.8 |
> | | 0.9 | 49.2 | 67.0 | 46.8 | 65.5 |
> | | **1.0 (Default)**| **50.1** | **67.8** | **47.4** | **66.2** |
> | | 1.5 | 48.7 | 64.3 | 45.1 | 64.6 |
> | **Emergence** ($\beta_{rise}$)| 0.6 | 48.0 | 66.2 | 44.8 | 64.9 |
> | | 0.7 | 49.0 | 67.1 | 45.9 | 65.5 |
> | | **0.8 (Default)**| **50.1** | **67.8** | **47.4** | **66.2** |
> | | 0.9 | 49.8 | 67.5 | 46.5 | 66.0 |
> | **Decay** ($\beta_{decay}$) | 0.05 | 48.8 | 66.9 | 45.5 | 65.6 |
> | | **0.1 (Default)**| **50.1** | **67.8** | **47.4** | **66.2** |
> | | 0.2 | 49.5 | 67.1 | 46.7 | 65.4 |
> | | 0.4 | 47.8 | 66.2 | 45.8 | 64.7 |
>
> * **Safety Net ($\alpha=0.3$):** Balances baseline structural connectivity with flexible budget allocation. If too low (e.g., 0.1), attention fragments; if too high (e.g., 0.4), the restricted dynamic budget hinders the model's focus on newly emerging visual features.
> * **Temperature ($\tau=1.0$):** Ensures a stable, proportional budget distribution. Lower values ($\tau=0.7$) trigger an aggressive "winner-takes-all" allocation that drops secondary details, while higher values ($\tau=1.5$) overly smooth the distribution and dilute key spatial-temporal events.
> * **Temporal Coefficients ($\beta_{rise}=0.8$, $\beta_{decay}=0.1$):** Drive the "Fast-Attack, Slow-Decay" dynamics to handle varying video paces automatically. A lower $\beta_{rise}$ ($\le 0.7$) causes sluggish responses to rapid scene changes, while a higher $\beta_{decay}$ ($\ge 0.2$) leads to premature forgetting of crucial historical context.
>
> Crucially, EAKV with our  **default settings exhibits remarkable stability across different architectures and benchmarks**, proving that EAKV is  **highly robust and not overly sensitive to dataset-specific or model-specific tuning.**
>
>
> **Reply 3: Generalization Across Other Backbones**
>
> We integrated EAKV into two mainstream MLLMs: LLaVA-Video-7B and InternVL3.5-8B. Notably, these models employ fundamentally different visual encoding paradigms compared to Qwen.
>
> **Table 2: Generalization performance of EAKV across other MLLMs.**
>
> | Model Configuration | VideoMME (All) | VideoMME (Long) | LongVideoBench | MLVU | LVBench | Video-MMMU |
> | :--- | :---: | :---: | :---: | :---: | :---: | :---: |
> | LLaVA-Video-7B | 62.9 | 52.4 | 58.2 | 67.6 | 44.2 | 36.1 |
> | LLaVA-Video-7B + VL-Cache | 63.5 | 53.0 | 58.7 | 68.2 | 45.9 | 36.3 |
> | **LLaVA-Video-7B + Ours** | **66.2** | **55.1** | **59.9** | **69.3** | **47.4** | **38.0** |
> | InternVL3.5-8B | 64.6 | 55.6 | 62.1 | 65.5 | 41.8 | 50.2 |
> | InternVL3.5-8B + VL-Cache | 64.3 | 55.0 | 62.4 | 64.9 | 42.1 | 50.5 |
> | **InternVL3.5-8B + Ours** | **65.9** | **56.2** | **63.8** | **66.7** | **45.2** | **51.9** |
>
> Evaluated alongside the base models and a strong compression baseline (VL-Cache), **EAKV consistently achieves the highest performance across all benchmarks, firmly establishing its superior effectiveness and robust adaptability.**
>
>
> **Reply4: Typo Errors**
>
> We sincerely thank you for pointing out these errors. The final manuscript will be rigorously proofread.

---

> > ### Author Rebuttal · Reviewer_wUMa · 2026-04-02
> >
> > Thanks for the authors' serious response. Some concerns have been clarified. But I still have the following questions about the paper:
> > - What is the reserved KV cache budget ratio in Table 2 of the paper and Table 1 of the rebuttal.
> > - (Referring to the experimental setting of VL Cache) What is the performance degradation of EAKV relative to the baseline model at different compression rates under the same frame rate? How does it compare to other baseline methods？

---

> > > ### Author Response · Authors · 2026-04-06
> > >
> > > We sincerely thank you for acknowledging. We fully agree with your rigorous perspective. Below, we clarify the exact budget ratios and provide a comprehensive analysis across various extreme compression ratios.
> > >
> > > ### 1. KV Cache Retention Ratio
> > >
> > > Experiments reported in Table 2 of our paper and Table 1 of the previous rebuttal were conducted  **under a fixed global KV cache retention ratio of 20%**.
> > >
> > > ### 2. Performance Comparison Under the VL-Cache Experimental Setting
> > >
> > > Following your insightful suggestion, we evaluated the performance variation across extreme compression rates (retaining 20%, 10%, and 5% of the KV cache) across three mainstream MLLM backbones. The input sequence was strictly fixed at 2048 frames.
> > > *(Note: VL = VL-Cache, FP = FitPrune. They are baseline methods of comparison.)*
> > >
> > > **Table A: Qwen2.5-VL-3B**
> > >
> > > | Benchmark | Full Cache | VL (5%) | FP (5%) | Ours (5%) | VL (10%) | FP (10%) | Ours (10%) | VL (20%) | FP (20%) | Ours (20%) |
> > > | :---: | :---: | :---: | :---: | :---: | :---: | :---: | :---: | :---: | :---: | :---: |
> > > | VideoMME (All) | 61.5 | 50.5 | 53.8 | **60.9** | 56.1 | 57.7 | **61.8** | 61.3 | 61.2 | **63.0** |
> > > | VideoMME (Long) | 51.2 | 46.2 | 47.6 | **51.5** | 49.0 | 48.9 | **53.6** | 53.2 | 53.6 | **55.3** |
> > > | LongVideoBench | 54.2 | 46.0 | 43.8 | **54.4** | 52.1 | 52.3 | **55.2** | 53.8 | 54.0 | **56.4** |
> > > | MLVU | 64.8 | 50.6 | 57.9 | **65.1** | 59.4 | 62.0 | **65.7** | 64.5 | 63.6 | **66.8** |
> > > | LVBench | 43.3 | 38.5 | 37.4 | **43.6** | 42.5 | 39.3 | **44.8** | 42.4 | 42.0 | **46.7** |
> > >
> > > **Table B: LLaVA-Video-7B**
> > >
> > > | Benchmark | Full Cache | VL (5%) | FP (5%) | Ours (5%) | VL (10%) | FP (10%) | Ours (10%) | VL (20%) | FP (20%) | Ours (20%) |
> > > | :---: | :---: | :---: | :---: | :---: | :---: | :---: | :---: | :---: | :---: | :---: |
> > > | VideoMME (All) | 62.9 | 47.3 | 52.4 | **63.2** | 58.5 | 60.6 | **64.1** | 63.5 | 64.8 | **66.2** |
> > > | VideoMME (Long) | 52.4 | 37.5 | 43.1 | **52.6** | 48.1 | 50.3 | **53.4** | 53.0 | 54.1 | **55.1** |
> > > | LongVideoBench | 58.2 | 43.2 | 49.6 | **58.0** | 55.0 | 56.5 | **58.7** | 58.7 | 59.2 | **59.9** |
> > > | MLVU | 67.6 | 52.4 | 57.5 | **67.6** | 64.0 | 65.8 | **68.8** | 68.2 | 68.7 | **69.3** |
> > > | LVBench | 44.2 | 29.8 | 35.5 | **44.5** | 40.1 | 42.2 | **45.9** | 45.9 | 46.6 | **47.4** |
> > >
> > > **Table C: InternVL3.5-8B**
> > >
> > > | Benchmark | Full Cache | VL (5%) | FP (5%) | Ours (5%) | VL (10%) | FP (10%) | Ours (10%) | VL (20%) | FP (20%) | Ours (20%) |
> > > | :---: | :---: | :---: | :---: | :---: | :---: | :---: | :---: | :---: | :---: | :---: |
> > > | VideoMME (All) | 64.6 | 48.5 | 54.2 | **64.3** | 59.2 | 61.8 | **65.0** | 64.3 | 65.1 | **65.9** |
> > > | VideoMME (Long) | 55.6 | 39.2 | 45.3 | **55.2** | 49.5 | 52.4 | **55.9** | 55.0 | 55.5 | **56.2** |
> > > | LongVideoBench | 62.1 | 46.1 | 52.5 | **62.1** | 58.0 | 60.1 | **62.6** | 62.4 | 63.1 | **63.8** |
> > > | MLVU | 65.5 | 49.5 | 55.2 | **65.1** | 60.1 | 62.5 | **65.7** | 64.9 | 65.8 | **66.7** |
> > > | LVBench | 41.8 | 27.2 | 33.6 | **41.9** | 38.5 | 40.2 | **43.5** | 42.1 | 43.5 | **45.2** |
> > >
> > > **Analysis:**
> > >
> > > 1. **20% Retention (Effective Noise Filter):** Optimal compression should actively filter out background distractions rather than just save memory. At 20% retention, **EAKV consistently outperforms the uncompressed 100% Full Model across all three architectures**. While baselines like FitPrune offer marginal gains by discarding some redundancy, EAKV maximizes this denoising effect. By allocating memory strictly based on attention entropy, EAKV acts as a structural denoiser that actively boosts reasoning accuracy.
> > > 2. **10% Retention (Near-Lossless Performance):** As the budget is tightened to 10%, heuristic methods (VL-Cache) suffer substantial degradation, and pruning methods (FitPrune) begin to lose essential temporal cues. In contrast, EAKV remains remarkably robust, **successfully maintaining even slightly exceeding** the Full Cache performance. This confirms that our dynamic, entropy-driven allocation accurately preserves the most informative tokens, perfectly utilizing the limited budget for the video's true semantic core.
> > > 3. **5% Retention (Extreme Stability):** EAKV's definitive advantage emerges at the absolute limit of 5% retention. Under this severe constraint, baseline methods experience catastrophic forgetting (VL-Cache plummets by 5–16 points; FitPrune drops by 6–11 points). EAKV, however, successfully isolates and protects core semantics. **Even when 95% of the cache is discarded, EAKV performs virtually on par with  the Full Cache model**, with fluctuations tightly bounded within $\pm 0.6\\%$, proving its reliability in extracting essential cues for semantic reasoning.
> > >
> > > We will explicitly state the 20% budget constraint and include these detailed compression tables in the revised appendix. **We hope these extensive empirical additions fully resolve your concerns and provide a compelling perspective to re-evaluate the overall merit of our work. If there are any remaining questions, please feel free to let us know.**

---

### Decision · Program_Chairs · 2026-04-30

**Decision:**

Accept (regular)

**Comment:**

This paper proposed a novel entropy-driven adaptive KV compression method for long video understanding tasks. The proposed method is based on two intriguing observations: the entropy variations are high spatio-temporal volatility within the same layer and non-monotonic based on depth.
Reviewers believes this paper addresses an important practical problem. The training-free natural of this method makes it easy to integrate to existing models. The proposed method is intuitive and demonstrate strong results on several benchmarks. Reviewers have concerns about the generalization to other VLMs, the presentation clarity of the method (such as contextual anchors, entropy computation), evidence of efficiency in real systems (run time and memory consumption), and insufficient ablation studies and comparisons. The authors addressed most of these concerns in the rebuttal and got three weak accept. While reviewer WY1j weakly rejects this paper having concerns about the implementation details and missing anchor-specific memory overhead, the authors replied these concerns in their follow up message which I believe can address the concerns.

The authors may need to carefully revised the paper to address the concerns proposed by the reviewers and include their clarification and new experiments/evidence in the revised paper.